# Time-varying stimuli that prolong IKK activation promote nuclear remodeling and mechanistic switching of NF-κB dynamics

Steven W. Smeal [1], Chaitanya S. Mokashi [1,4], A. Hyun Kim [1], P. Murdo Chiknas[1] & Robin E. C. Lee [1,2,3] ✉

Temporal properties of molecules within signaling networks, such as subcellular changes in protein abundance, encode information that mediate cellular responses to stimuli. How dynamic signals relay and process information is a critical gap in understanding cellular behaviors. In this work, we investigate transmission of information about changing extracellular cytokine concentrations from receptor-level supramolecular assemblies of IKK kinases downstream to the NF-κB transcription factor. In a custom robot-controlled microfluidic cell culture, we simultaneously measure input-output encoding of IKK-NF-κB in dual fluorescent-reporter cells. When compared with single cytokine pulses, dose-conserving pulse trains prolong IKK assemblies and lead to disproportionately enhanced retention of nuclear NF-κB. Using particle swarm optimization, we demonstrate that a mechanistic model does not recapitulate this emergent property. By contrast, invoking mechanisms for NF-κB-dependent chromatin remodeling to the model recapitulates experiments, showing how temporal dosing that prolongs IKK assemblies facilitates switching to permissive chromatin that sequesters nuclear NF-κB. Remarkably, using simulations to resolve single-cell receptor data accurately predicts same-cell NF-κB time courses for more than 80% of our single cell trajectories. Our data and simulations therefore suggest that cell-to-cell heterogeneity in cytokine responses are predominantly due to mechanisms at the level receptor-associated protein complexes.

The typical body of an adult human has 10 s of trillions of cells[1], and in some cases, individual cell behaviors can affect the entire organism. When cells are exposed to different biologic cues in their microenvironment, such as inflammatory cytokines, they activate dynamic signal transduction networks that mediate vital cell fate decisions. Deregulation of these networks contributes to a panoply of autoimmune diseases and cancers[2–4]. Variability and nonlinearity are typical characteristics of cellular signal transduction that limit our understanding of healthy and diseased cell behaviors and decision

processes. Cell-to-cell variability within a clonal populations exposed to the same conditions arises from intrinsic and extrinsic sources of biochemical noise associated with stochastic molecular interactions that impact transcription rates, protein expression, among other biological processes[5–8]. Meanwhile, nonlinearity emerges due to complex biomolecular networks that include a host of feedback and feedforward loops. Adding to this complexity, the extracellular environment is dynamic, and cells are constantly exposed to multiple signaling molecules with changing concentrations. The combined effects of

[1]Department of Computational and Systems Biology, School of Medicine, University of Pittsburgh, Pittsburgh, PA 15213, USA. [2]Center for Systems Immunology, School of Medicine, University of Pittsburgh, Pittsburgh, PA 15213, USA. [3]Department of Physics and Astronomy, University of Pittsburgh, Pittsburgh, PA 15260, USA. [4]Present address: Altos Labs, Redwood City, CA 94065, USA. ✉e-mail: robinlee@pitt.edu

variability, nonlinearity, and dynamic input signals contribute to versatility of signaling pathways to regulate a wide range of responses. Therefore, experiments that combine single cell dynamics with computational modeling are important to reveal the capabilities of signaling systems and understand their emergent properties[9].

When inflammatory cytokines in the extracellular milieu activate transmembrane cytokine receptors, clustered receptors recruit cytoplasmic adaptors and enzymes to form a supramolecular protein assembly near the plasma membrane[10,11]. As the assembly matures, polyubiquitin scaffolds form around the protein core. A hallmark of cytokine responses is recruitment-based activation of IκB kinase (IKK) complexes to assemblies through the ubiquitin binding domain of NEMO, the IKK regulatory subunit[12–15]. Using fluorescent protein (FP) fusions of NEMO, individual assemblies can be visualized by live-cell microscopy as diffraction-limited puncta near the plasma membrane[16,17]. The mature protein assembly is a signal integration hub that activates IKK to coordinate downstream inflammation-driven NF-κB signaling. Although the supramolecular assemblies, referred to as 'complex I' (CI)[10,11], were first characterized for responses to tumor necrosis factor (TNF), interleukin-1ß (IL-1) and other cytokines produce CI-like assemblies with cytokine-specific receptors and adaptor proteins. CI and CI-like assemblies are reliant on different combinations of ubiquitin chain scaffolds[15,16,18]. However, all regulate IKK activation via induced proximity of IKK with other signaling mediators that reside on the assembly[19]. For simplicity, we refer to the family of CI and CI-like assemblies simply as 'CI'.

IKK activity following induction of CI promotes degradation of NF-κB inhibitor proteins (IκB) in the cytoplasm, and nuclear accumulation of nuclear factor κB (NF-κB) transcription factors[20]. Temporal properties of NF-κB in the nucleus encodes a dynamic transcriptional signal, regulating diverse gene expression programs that promote cell survival and propagate inflammatory signals. Live-cell tracking of FP fusions of the RelA subunit of NF-κB have provided instrumental data to understand and model transcriptional mechanisms and emergent properties that place the NF-κB pathway among exemplars of dynamical biological systems[21–23]. Key to these discoveries are negative feedback mediators, particularly IκBα, that are transcriptionally regulated by NF-κB. Following NF-κB activation, newly synthesized IκBα promotes nuclear export and cytoplasmic sequestration of NF-κB, restoring its baseline cytoplasmic localization. Combined experiments and models have revealed mechanisms for dynamical regulation of nuclear NF-κB and transcriptional feedback. For example, the temporal dynamics of nuclear NF-κB, including both RelA and c-Rel, have been shown using information theory frameworks to be *de facto* carriers of information, encoding and decoding aspects of cytokine type and abundance from the milieu[24–28]. Together these works suggest that NF-κB is a regulator for several distinct transcriptional response programs. However, we have limited understanding upstream of NF-κB for how dynamic extracellular signals are perceived by the cell and transduced through IKK recruitment at CI-like assemblies.

Typically, cellular signaling pathways have been studied using dose-response approaches that expose cells to continuous and unchanging extracellular stimuli. While responses to static concentrations provide a foundational readout for investigating cell behaviors, they are neither comparable to the dynamic signals observed in vivo nor sufficient to fully probe the versatility of cellular responses. We and others have recently shown that dynamic cell cultures can be achieved using PDMS- or acrylic-based microfluidic flow devices that vary with trade-offs between complexity, precision, and multiplex capabilities[29–32]. In most cases, single cell studies of signal transduction in dynamic microenvironments have revealed unexpected emergent properties that have inspired significant refinement of models that aim to predict and understand the underlying biological circuits[29,33–36].

In this work, we investigate IL-1-induced signaling in U2OS osteosarcoma cells to understand how information for time-varying extracellular cytokines is encoded within cells. U2OS cells provide a robust human model for studying polyubiquitin-dependent mechanisms of IKK regulation and nuclear NF-κB dynamics in response to cytokines[16–18]. The dual reporter cells were CRISPR-modified for endogenous FP fusions to observe active receptor complexes through supramolecular assemblies of EGFP-NEMO, the regulatory IKK subunit, and downstream dynamics of an mCherry-fusion of the NF-κB RelA subunit. Using live-cell imaging in a custom robot-controlled microfluidic cell culture system, we simultaneously measure upstream and downstream reporters in single cells exposed to dynamic stimuli. By observing input-output (I/O) encoding of IKK-NF-κB signals and comparing cells exposed to a single cytokine pulse with varied duration, we show monotonicity between the aggregate of activated receptor complexes and downstream transcription factor (TF) dynamics. This result is consistent with our previous investigation of cells exposed continuously to static cytokine concentrations[17]. Remarkably, monotonicity of I/O encoding breaks down when cells are exposed to a series of short cytokine pulses. We observe that dynamic TF responses to pulse trains are significantly greater than expected from a single pulse, even when pulse trains are compared to a bolus of a larger overall cytokine exposure. Enhanced TF responses to a cytokine pulse train can be attributed to marked alteration of nuclear export dynamics of NF-κB, transitioning from first-order to sustained zero-order kinetics in cells where activated receptor complexes persists for longer than 80 minutes. Using particle swarm optimization, we demonstrate that a mechanistic model does not recapitulate this emergent property of the IKK-NF-κB signaling axis and fails to switch to zero-order kinetics in response to a pulse train. By contrast, adding mechanisms for DNA binding and NF-κB-dependent chromatin remodeling, the resulting models recapitulate all the experimental findings. Our model and validation experiments suggests that temporal dosing that prolongs IKK activation facilitates pseudo-zero-order switching by two coupled mechanisms that prolong nuclear NF-κB retention without requiring persistent or saturating extracellular cytokine conditions: i) by promoting permissive chromatin states that expose new NF-κB binding sites; and consequently, ii) by reducing NF-κB nuclear export through increased push-pull competition for binding between generic and promoter-specific protein-DNA interactions versus protein-protein interactions with negative feedback mediators. Furthermore, simulations using the calibrated model to resolve experimental CI time courses from a complete set of single-cell validation data that were not used during optimization, accurately predicts same cell NF-κB responses for over 80% of single-cell time courses. Together, our results demonstrate that when overall cytokine dosage is limited, temporal stimuli can encode distinct cellular behaviors, and that cell-to-cell variability in cytokine responses is largely accounted for by variability of receptor-level mechanisms.

## Results

### IL-1 pulse duration produces a monotonic correlation between same-cell IKK and NF-κB responses

We previously developed a CRISPR-modified U2OS cell line that co-express FP fusions of NEMO (EGFP-NEMO) and RelA (mCh-RelA) from their endogenous loci. These dual-reporter cells were used to investigate the IKK-NF-κB signaling axis in cells exposed to continuous stimuli[17]. By simultaneously measuring time courses of fluorescence intensity for supramolecular assemblies of EGFP-NEMO as a reporter for CI and nuclear NF-κB in the same cell, these studies revealed fundamental aspects of signal encoding in the pathway. We reported that extracellular cytokine concentrations control the numbers and timing of formation for CI assemblies. Experiments that tracked EGFP-NEMO

at individual CI assemblies revealed that fluorescence intensity time-courses of single puncta are invariant between different cytokine concentrations. Further experiments revealed cytokine-specific encoding, where a quantized number of EGFP-NEMO molecules is recruited at each CI structure. Here, TNF-induced receptor assemblies are shorter-lived and recruit approximately 30% the peak amount of EGFP-NEMO when compared with brighter and longer-lived IL-1-induced assemblies. Finally, we observed that activity from CI puncta is pooled for downstream signal transmission where the aggregate area under the curve (AUC) of CI puncta numbers is a strong same-cell predictor of the AUC nuclear NF-κB response.

Our previous results helped define the rules of signal encoding in terms of cytokine-specificity and static cytokine concentrations. However, these experiments did not investigate the orthogonal axis of how time-varying patterns of cytokines influence IKK-NF-κB signaling, where studies have shown that NF-κB activation depends not only on ligand dose but also on the duration of stimulation[29,37,38]. We therefore set out to probe the pathway by first measuring the dynamics of IKK and NF-κB responses as a function of cytokine pulse duration. To generate cytokine pulses while simultaneously performing high-magnification single-complex imaging, we leveraged a recently developed dynamic stimulation system (DSS). The DSS consists of a custom robotic gravity pump controller that coordinates laminar fluid streams in a paired microfluidic cell culture device[30]. The microfluidic device was simplified from the previous DSS instrumentation as a two-inlet cell culture chamber (Fig. 1a, Supplementary Fig. 1a, b, and Supplementary Data 1). Here, one inlet is attached to a fluid reservoir with a mixture of cytokine, indicator dye, and cell culture medium, and to the other, a reservoir with medium only. Using the gravity pump to vertically relocate the fluid reservoirs generates hydrostatic pressure differentials at the inlets of the microfluidic device and positions the laminar cytokine-containing stream over the cells within (Fig. 1b). By varying the relative heights of the cytokine-containing and medium-only reservoirs, the system can be used to generate pulses in the experimental region of the cell culture device (orange boxes, Fig. 1b). In principle, the automated system described here operates like a manually controlled system we described previously[29], but with enhanced precision, reproducibility, and constant flow rates[30].

Using our dual-reporter cell line, we investigated the same-cell dynamics of NEMO assemblies and NF-κB nuclear translocation in a microfluidic device. We focused on cellular responses to 10 ng/mL of IL-1, a non-saturating cytokine concentration that produces vibrant NEMO assemblies and unambiguous nuclear NF-κB responses[17]. Consistent with our previous results, dual-reporter cells in the device showed transient localization of EGFP-NEMO to CI-like complexes, peaking in numbers by approximately 20 minutes, and robust nuclear translocation of mCh-RelA (Fig. 1c, and Supplementary Movie 1). Next, we measured single cell time-courses of EGFP-NEMO puncta and nuclear mCh-RelA in response to a single IL-1 pulse across a range of durations. Previously, we showed that time-integrated AUCs for trajectories of NEMO spot numbers and nuclear RelA fold change are scalar descriptors of single cell time courses that encode the most information about cytokine concentration[28,39]. Consistent with these results, AUCs of EGFP-NEMO spot numbers and nuclear mCh-RelA in single cells show increasing responsiveness with pulse duration and form a strongly monotonic continuum of same-cell correlations (Fig. 1d–f).

Previous works have shown that mammalian cell lines may follow an 'area-rule', where the fraction of responsive cells and overall NF-κB activation strength is proportional to the product of concentration and duration of cytokine stimulation[29,38]. Our current results are consistent with the 'area-rule', showing that information about extracellular pulse duration is also well determined when observed upstream via EGFP-NEMO puncta that provide a readout for CI

activation. Next, we asked if the area rule remains true for increasingly complex patterns of dynamic cytokine stimulation.

## Cytokine pulse trains produce significantly greater NF-κB responses than expected from a single pulse

Unlike most secreted proteins, IL-1β does not follow the conventional ER-Golgi pathway[40], instead, it accumulates in the cytosol as an inactive precursor (pro-IL-1β). Several pathways mediate its activation and release, including inflammasome assembly, which rely on caspase-1-dependent maturation of pro-IL-1β and trigger its rapid secretion via secretory lysosomes, microvesicles, or pyroptosis[41]. In vivo, transient IL-1β exposure could arise from repeated bursts by individual cells or asynchronous activation across neighboring cells, resulting in brief waves of cytokine release[42]. Pulses of inflammatory cytokines have been shown to selectively regulate long lasting transcriptional programs. For example, in neuroblastoma cells, a single 5-minute pulse of TNF can initiate transcription of NF-κB-regulated early response genes that persist for hours, but not late response genes that require either constant or repeated cytokine stimulation[22]. The DSS provides an opportunity to test whether cytokine pulses regulate distinct dynamic intracellular signal transduction programs, and more generally, probe the limits of signaling. We therefore asked whether a succession of short cytokine pulses in the extracellular milieu is encoded distinctly by the IKK-NF-κB axis, or if responses to a pulse train follows an 'area rule' equivalent to a single bolus of same overall dose.

We compared cellular responses between a single 6-minute pulse that produces an intermediate response strength (Fig. 1c and d), and multiple shorter pulses with the same cumulative IL-1 exposure but spread out in time. We partitioned a single 6-minute pulse equally into two, three, or four short pulses, and compared inter-pulse gap durations between 5 and 20 minutes (Fig. 2 and Supplementary Fig. 2). To verify that the microfluidic chip conserved the cumulative cytokine exposure between conditions, we performed high-temporal-resolution experiments with a fluorescent dye. Experiments comparing the single 6-minute and the four 1.5-minute pulses revealed that the AUCs for both stimuli are reproducible and conserve the total cytokine exposure (Supplementary Fig. 1c, d), as expected from previous calibration experiments[30]. In comparison with a single 6-minute pulse, time-courses for NEMO puncta numbers in response to pulse trains showed lower peak values and longer spot persistence (Fig. 2a). Although in some pulse train conditions there was a subtle overall increase in AUC of NEMO puncta, these values typically did not reach significance and remained smaller than exposure to a larger single 15-minute pulse (Fig. 2 and Supplementary Fig. 2). Remarkably, despite only subtle and often non-significant increases in the AUC of IKK responses, many of the pulse-train conditions showed significant increases in the AUC of nuclear mCh-RelA time-courses (Fig. 2b). Even though pulse trains produced comparable peak amounts of nuclear RelA with similar timing to a dose-conserving single bolus, contributing to the stronger NF-κB AUC response was markedly slower nuclear export. Nuclear export of NF-κB following a pulse train appeared almost linear, following zero-order kinetics where rates are independent of reactant concentration. Nuclear export of RelA in response to a pulse train contrasts with most single pulse responses less than 30 minutes, which exhibit exponential nuclear export, as expected from first- and higher-order biomolecular reactions (Fig. 2, and Supplementary Movie 2). The enhancement effect of linear nuclear RelA export is exemplified by the condition with four 1.5-minute pulses of IL-1 separated by a 5-minute gap (4×1.5), which showed a greater than 2-fold increase in overall AUC. The nuclear RelA response strength for the 4×1.5 condition even surpassed responses to a 15-minute pulse despite the former having less than half the total cytokine exposure (Fig. 2b and Supplementary Movie 3). Finally, the monotonicity of same-cell correlations between AUCs of NEMO spots and nuclear RelA was disrupted for pulse train conditions (Fig. 2c).

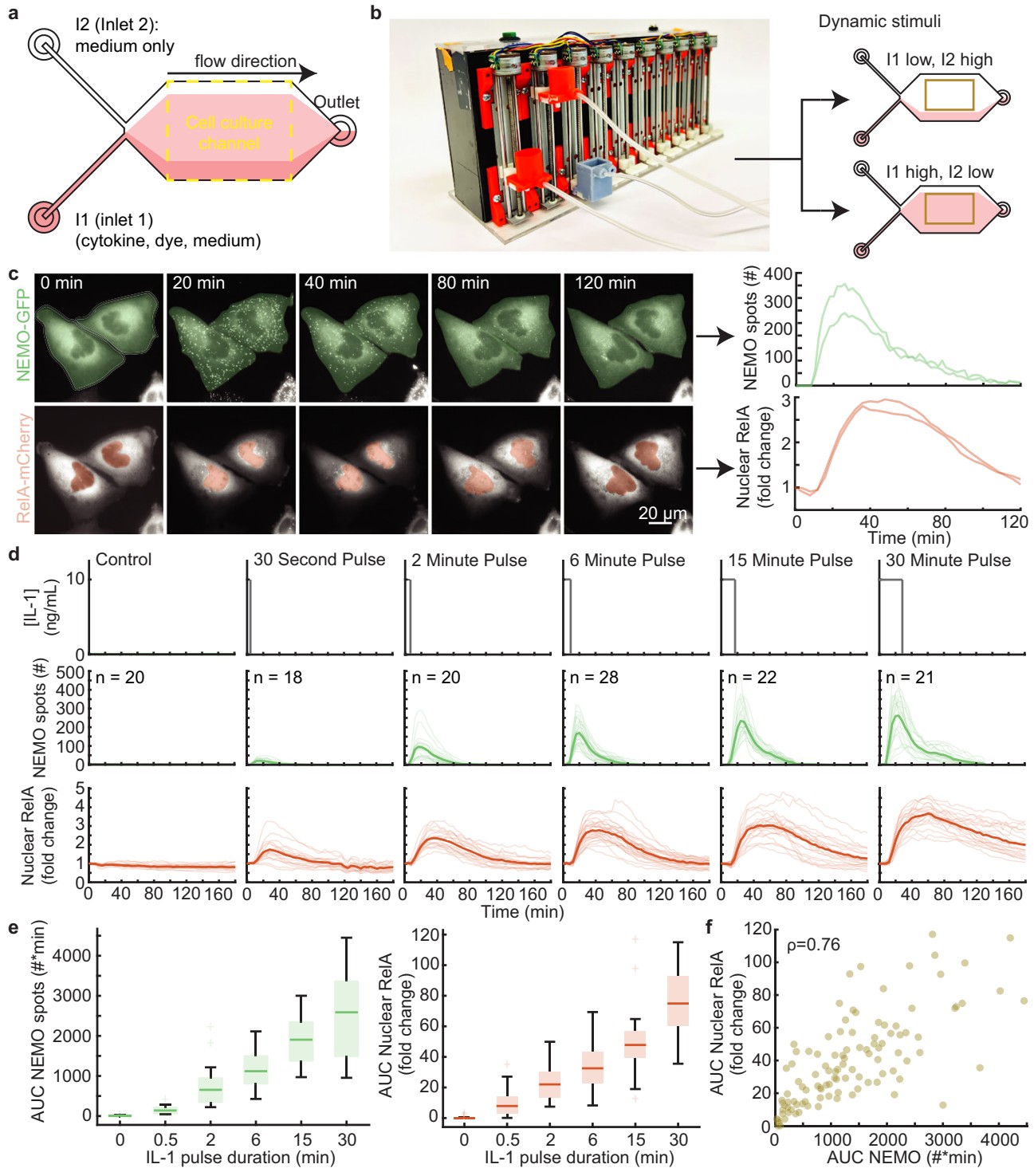

The unexpected increase in the response strength of nuclear RelA to a pulse train is an emergent property of the signaling pathway. We next asked whether the emergent property is specific to cytokine or cell type and found that strongly TNF-responsive KYM-1 rhabdomyosarcoma cells that express FP-RelA[28,29] show similarly enhanced responses to TNF pulse trains (Supplementary Fig. 3). Notably, some pulse train conditions diminish the emergent property effect. For example, conditions with fewer pulse numbers and longer inter-pulse gap lengths in the order of 10-20 minutes tend to have weaker nuclear RelA responses for both cell types and both cytokines (Supplementary Figs. 2 and 3). Together, these results suggest that cells can bypass the area rule in response to dynamic milieus and encode more information

through CI assemblies than is represented by the scalar AUC of NEMO puncta.

## Stimulation patterns that prolong NEMO puncta shift the mechanism of NF-κB nuclear export

Since the AUC of NEMO puncta could not explain enhanced nuclear NF-κB in response to a pulse train, we asked if other quantitative features of EGFP-NEMO single cell time courses reflect the emergent property. When temporal features were quantified from NEMO time courses, distributions for both the timing of maximal ($t_{max}$) and time to adaptation ($t_{adapt}$) of NEMO spot numbers showed trends like the AUC of nuclear RelA (Fig. 3a-c).

**Fig. 1 | NEMO features determine same-cell NF-κB responses to a single cytokine pulse. a** Schematic of a two-inlet, one outlet microfluidic chip with fluid flow from left to right. Inlet 1 (I1) contains a mixture of cytokine, dye, and medium, while inlet 2 (I2) contains only media. The fluorescent dye is used exclusively to establish the position of the cytokine-containing stream. U2OS cells are cultured and imaged within the central cell culture channel. On average, 22 cells were imaged for each condition. **b** Image of the custom gravity pump. Inlet reservoirs I1 and I2 are connected to the orange basins and the outlet connected to the gray basin. The gravity pump dynamically adjusts the relative heights of the basins to control fluid flow. When the I1 basin is positioned lower than I2, cells are exposed to media only (top) and when I1 is higher, cells are exposed to cytokine (bottom). **c** Time-lapse images of CRISPR-modified U2OS cells lines containing fluorescent protein fusions from their endogenous loci (left; see also Supplementary Movie 1). Quantification of upstream activated receptor complex measured through the formation and tracking of single EGFP-NEMO puncta (top). Quantification of downstream activation of the NF-κB (RelA) transcriptional system calculated by the fold change of mCh-RelA fluorescence in the nucleus (bottom). **d** Schematic

illustrating the IL-1 stimulation profile applied to cells using the microfluidic device (top). Single cell time-courses for EGFP-NEMO spot numbers (green, middle) and mCh-RelA nuclear fold change (red, bottom) in response to a single pulse of IL-1 at 10 ng/ml with varying pulse durations. Light-colored trajectories represent individual cells, while bold trajectories represent the average response. **e** The area under the curve (AUC) for the number of IL-1 induced spots (green, left) and the fold change of nuclear RelA (red, right) increases with pulse duration. Single cell n values are listed in panel (**d**). **f** Same-cell correlations from dual-reporter cells show the aggregate AUC of activated receptor complexes and downstream transcriptional activity display a monotonic relationship for cells exposed to IL-1 pulses (Spearman ρ = 0.76). The unstimulated cells were not included in the correlation and plot. The scatterplot has $n = 109$ same-cell data points. For boxplots, the boxes represent the interquartile range with the median indicated as the center line. The bounds of the box correspond to the 25th and 75th percentiles. Whiskers extend to the minima and maxima values within 1.5 interquartile range, and points beyond are shown as outliers. Source data are provided as a Source Data file.

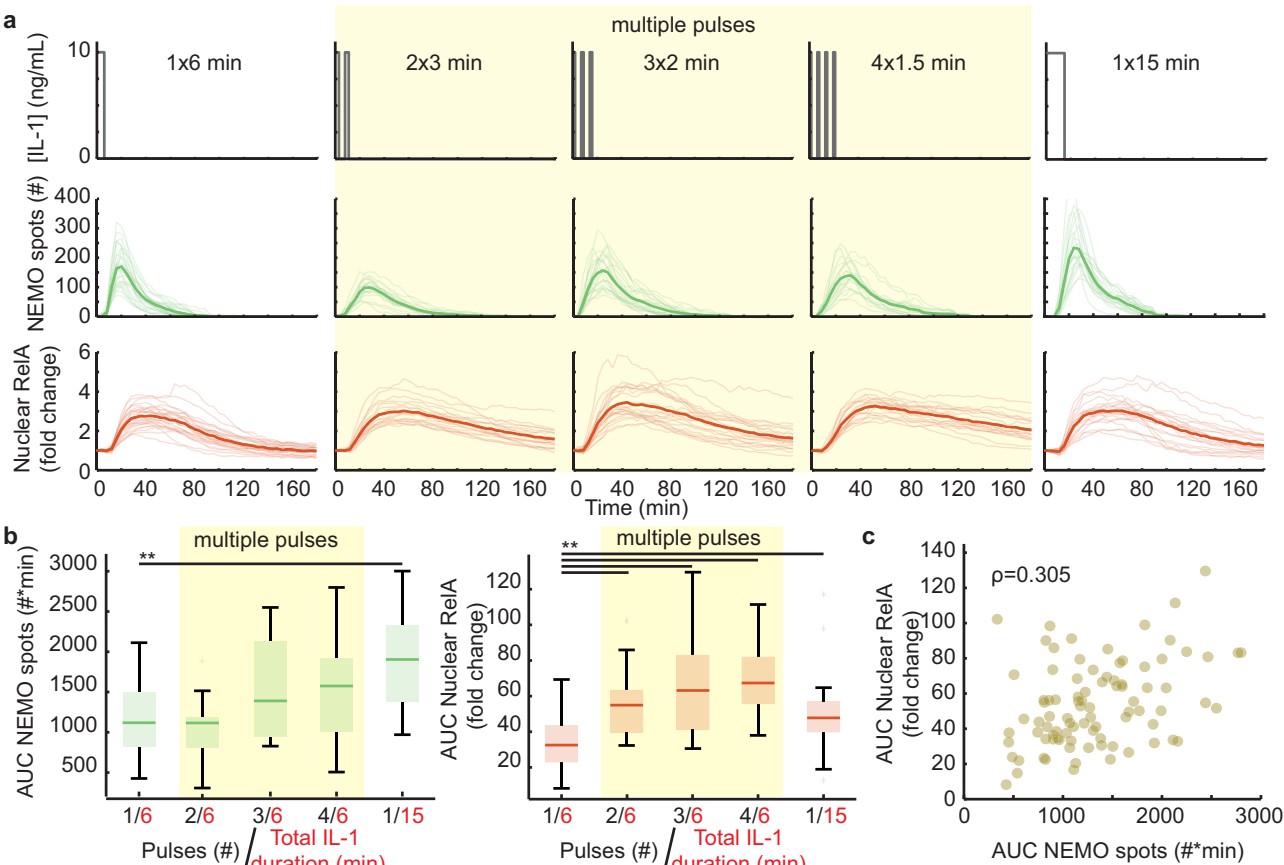

**Fig. 2 | NF-κB responses to a dose-conserving cytokine pulse train are enhanced. a** Schematic illustrating the IL-1 stimulation profile applied to cells using the microfluidic device (top). Single cell time-courses for EGFP-NEMO spot numbers (green, middle) and mCh-RelA nuclear fold change (red, bottom) in response to multiple short pulses. Single 6- and 15-minute pulses are shown as reference points, representing equivalent and greater than twice the total cytokine exposure, respectively, compared to the multiple short pulses (yellow highlighted conditions; see also Supplementary Movies 1 and 2). On average, 20 single cells were imaged per condition. **b** The area under the curve (AUC) analysis of NEMO spot numbers (green, left) shows no significant difference between single and multiple pulse stimulation patterns. For comparison, the single 6- and 15-minute pulses do show a significant difference (left). By contrast, the AUC for nuclear RelA fold change

indicates a significant difference between single and multiple pulse stimulation patterns (red, right). Median and interquartile ranges are shown. Student's two-sided t-test p-values for pairwise comparisons are: 1/6 vs 1/15 = $10^{-5}$ for NEMO and 0.005 for RelA; 1/6 vs. 2/6 RelA = $10^{-4}$; 1/6 vs. 3/6 RelA = $10^{-5}$; 1/6 vs. 4/6 RelA = $10^{-8}$. Single cell n values are: 1/6 = 28; 2/6 = 23; 3/6 = 19; 4/6 = 19; 1/15 = 22. **c** The aggregate AUC of activated receptor complexes and downstream transcriptional activity is not monotonic when responses to multiple short pulses are combined. The scatterplot has $n = 89$ same-cell data points. For boxplots, the boxes represent the interquartile range with the median indicated as the center line. The bounds of the box correspond to the 25th and 75th percentiles. Whiskers extend to the minima and maxima values within 1.5 interquartile range, and points beyond are shown as outliers. Source data are provided as a Source Data file.

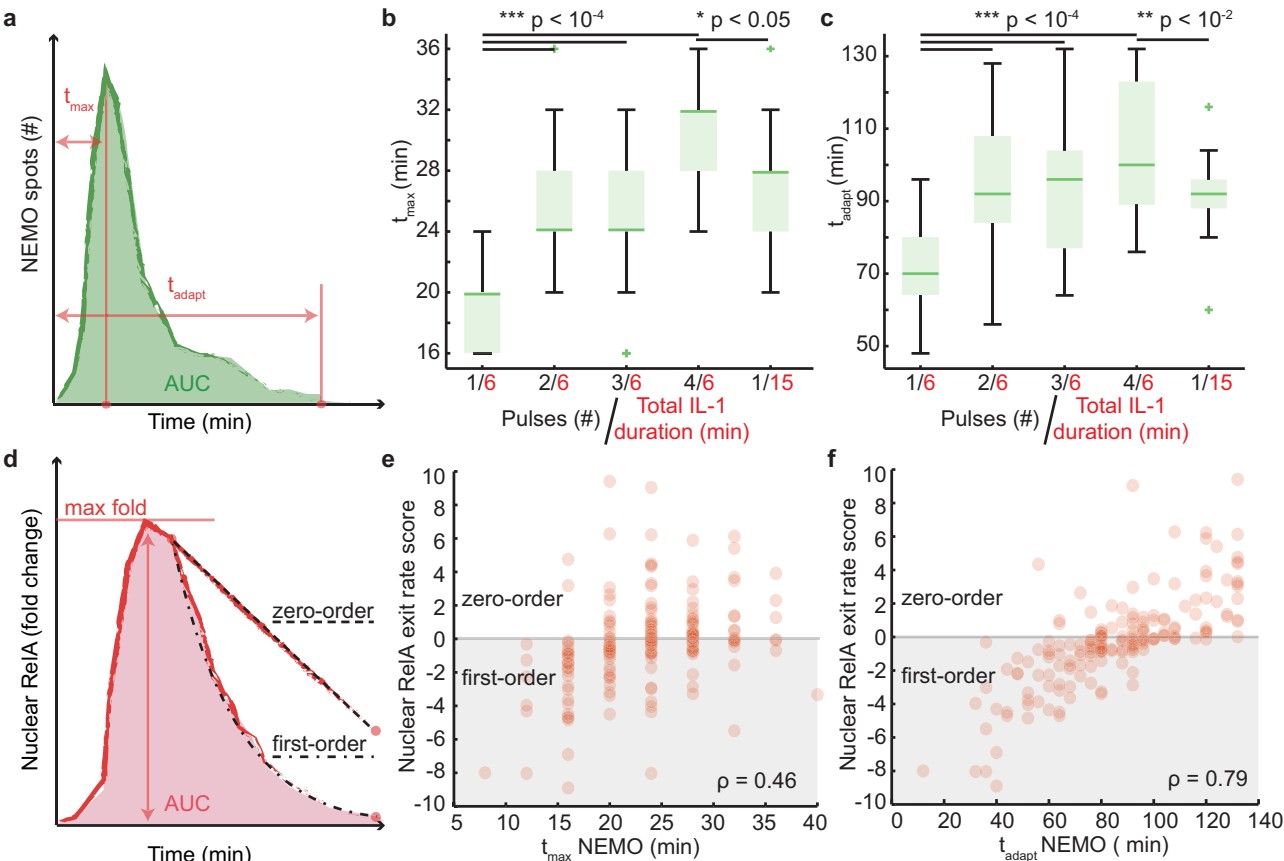

**Fig. 3 | Temporal features of NEMO spot numbers correlate with NF-κB and zero-order export kinetics following a cytokine pulse train. a** Schematic of a typical NEMO spot trajectory highlighting features for the 'time to maximum' ($t_{max}$) and 'adaptation time' ($t_{adapt}$). **b** Boxplots showing $t_{max}$ for single-cell trajectories under 6-minute single and multi-pulse stimulation. Student's two-sided t-test p-values for pairwise comparisons are: 1/6 vs. 2/6 = $10^{-8}$; 1/6 vs. 3/6 = $10^{-5}$; 1/6 vs. 4/6 = $10^{-14}$; 4/6 vs. 1/15 = 0.04. Single cell n values are: 1/6 = 28; 2/6 = 23; 3/6 = 19; 4/6 = 19; 1/15 = 22. **c** Boxplots showing $t_{adapt}$ for single-cell trajectories under 6-minute single and multi-pulse stimulation. Student's two-sided t-test p-values for pairwise comparisons are: 1/6 vs. 2/6 = $10^{-9}$; 1/6 vs. 3/6 = $10^{-5}$; 1/6 vs. 4/6 = $10^{-8}$; 4/6 vs. 1/15 = 0.04. **d** Schematic of a typical nuclear RelA fold change trajectory highlighting

the max fold change and the area under the curve (AUC). Additionally, examples for zero-order and first-order kinetics are illustrated for nuclear RelA export. **e** Same cell correlations for time-courses of nuclear RelA exit rates scores and the $t_{max}$ of NEMO. **f** Same cell correlations for time-courses of nuclear RelA exit rates scores and the $t_{adapt}$ of NEMO (min). For both scatterplots, n = 170 same-cell data points. For boxplots, the boxes represent the interquartile range with the median indicated as the center line. The bounds of the box correspond to the 25th and 75th percentiles. Whiskers extend to the minima and maxima values within 1.5 interquartile range, and points beyond are shown as outliers. Source data are provided as a Source Data file.

Next, we set out to establish quantitative features of EGFP-NEMO that correlate with features of nuclear RelA. Because the maximum of nuclear RelA fold change did not show a strong trend between pulse-trains and dose-conserving single-pulses, we focused on scoring rates of nuclear RelA export (Fig. 3d and Supplementary Fig. 4a). As mentioned earlier, we visually observed that dynamics for nuclear RelA export appeared to switch from a first order to a pseudo-zero-order reaction (Figs. 2a and 3d) in response to a pulse train. Here, we define pseudo-zero-order kinetics as nuclear RelA trajectories that exhibit two properties. First is the prolonged adaptation of nuclear RelA that is significantly longer than expected from the dose-conserved single pulse. Second is a nuclear export profile that resembles a linear rate (zero-order kinetics) - likely arising from the interplay of multiple first-order processes that collectively saturate or buffer mechanisms for export dynamics. To quantify this observation, single-cell nuclear RelA export dynamics were fitted to two models that represent zero-order and first-order kinetics (Supplementary Fig. 4b; and see methods). In the results, increasingly positive scores indicate a progressively stronger zero-order exit rate, while negative values correspond to first-order exit rates. We observed that single 6-minute and 15-minute pulse shows strong first-order dynamics and the responses to pulse trains showed strong zero-order kinetics (Supplementary Figs. 1e and 4c).

Extending imaging to 5 h revealed that nuclear RelA export after multi-pulse stimuli continues to follow pseudo-zero-order kinetics long after cells from a single 6-minute pulse adapt (Supplementary Fig. 4d, e), suggesting that a mechanistic shift in export dynamics has occurred. Finally, we correlated features of EGFP-NEMO against the nuclear RelA export order score. Although $t_{max}$ of NEMO only correlated weakly, by contrast, $t_{adapt}$ of NEMO showed a strong monotonic correlation with the nuclear RelA exit rate score (Fig. 3e, f). Based on our data, stimulation profiles that prolong CI assemblies beyond 80 minutes shift the nuclear export rate of RelA towards a pseudo-zero-order process and thus enhance the AUC of the nuclear NF-κB response.

## A model with chromatin remodeling recapitulates the emergent property and nuclear NF-κB export

Mechanistic models provide mathematical representations of biological systems, enabling researchers to test hypotheses and synthesize their understanding of system-level behaviors. Ordinary differential equation-based models of the NF-κB transcriptional system have been deeply informative. Previous models parameterized to quantitative assays and single cell data have been used to investigate nuclear translocation dynamics of NF-κB, emphasizing negative feedbacks from IκB isoforms[43] and A20[22], as well as mechanisms for fold change

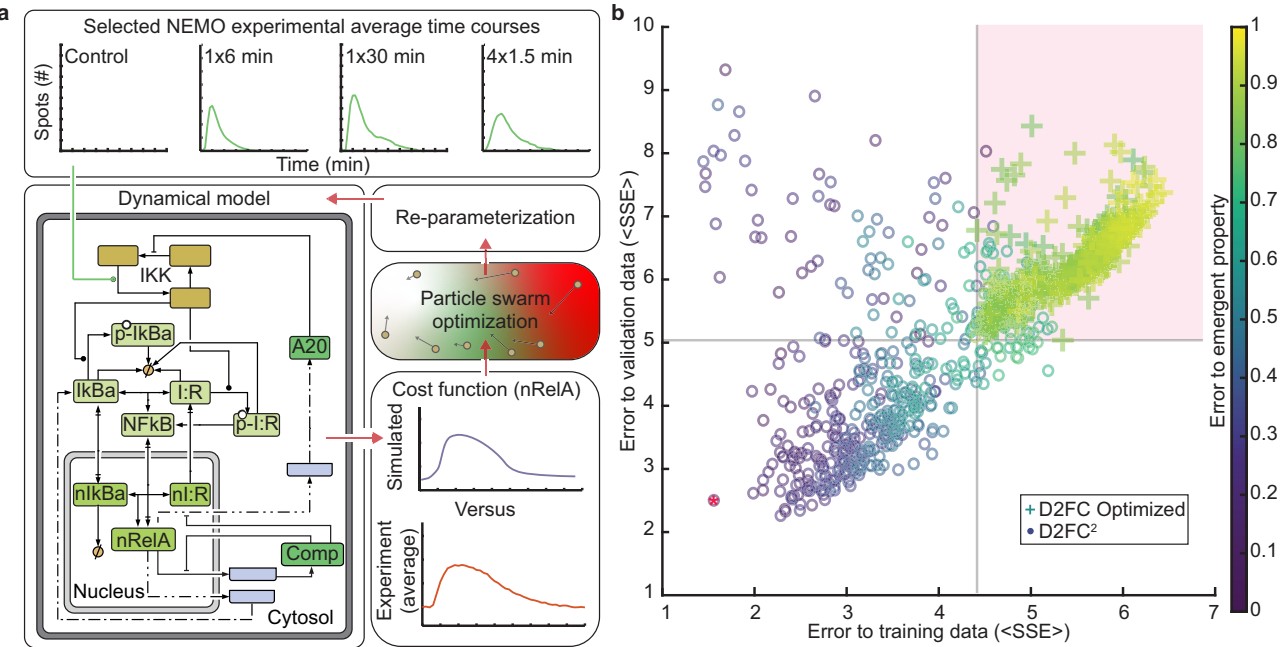

**Fig. 4 | A model with DNA-binding and chromatin remodeling recapitulates emergent properties of average cellular responses to dynamic stimuli.**
**a** Schematic of the mechanistic model fitting strategy. As input, the model uses experimental averages of EGFP-NEMO time-courses from four conditions, chosen to tune the model to the emergent property. Simulated NF-κB responses are compared against experimental results and used as a cost function for particle swarm optimization (PSO). **b** Results of PSO from 500 replicates for each model architecture. Each model parameterization is evaluated based on the sum of squared error (SSE) to the 4 training conditions (x-axis), to the SSE of averages from the 5 validation conditions not used for PSO (y-axis), and emergent property error score (colorbar). D2FC optimized models were limited to the quadrant highlighted in pink. The red star indicates the overall best D2FC$^2$ model. Source data are provided as a Source Data file.

detection (called D2FC, derived from previous models)[39]. Each of these models has been built upon their predecessors to answer different questions. However, none of the models have been parameterized to single receptor-complex data and dynamic stimuli.

To explore the possible biological mechanisms that contribute to zero-order switching and enhanced nuclear RelA retention in response to a pulse train, we first asked if the D2FC model directly recapitulates experiments. Since we have a compendium of single-cell data from dual-reporter cell lines, we modified the model to activate IKK directly from experimental time courses of EGFP-NEMO puncta and simulate nuclear NF-κB responses (Fig. 4a). We first tested the previously published parameterization of the D2FC model, using a fitted scaling parameter to interface experimental data for EGFP-NEMO puncta that produces a best fit simulated NF-κB response (see methods). Although the naive D2FC model replicates the AUC of NF-κB for responses to a single pulse, the model results did not show increased AUC of nuclear NF-κB nor switching to zero-order export mechanisms when simulations were run using EGFP-NEMO pulse-train data as inputs (Supplementary Fig. 5).

Next, we tested if the topology of the D2FC model can reproduce the emergent property using parameter optimization. Particle swarm optimization (PSO) is a bio-inspired algorithm that searches the parameter space for candidate solutions, using an objective function to measure the quality of fit for each[44]. Implementations of PSO often identify nearly global optimal parameterizations in high-dimensional and rugged objective landscapes. To decrease the computational time, we considered a minimal dataset of 4 experimental conditions for EGFP-NEMO that manifest the emergent property at the level of nuclear NF-κB dynamics (consisting of control, 1x6min, 4×1.5 min, and 1x30min pulse data; Fig. 4a top panel). Subsequently, we used PSO to simulate responses to the average of EGFP-NEMO time-courses from each condition, using the average of nuclear mCh-RelA for each condition as the objective function (Fig. 4a). PSO was repeated 500 times

to generate parameter sets that sample the model's best fits to data. For the 500 parameter sets, we performed post-hoc analysis to evaluate three error criteria for each: i) error in producing a nearly 2-fold increase in the AUC of nuclear NF-κB comparing 1x6min and 4×1.5 min conditions, referred to as the 'emergent property' score; ii) the average sum squared error ($<$SSE$>$) of fits to the 4 training conditions; and, (iii) the SSE of fits to validation data consisting of 5 mean experimental conditions that were not used for PSO (Fig. 4b). Remarkably, each of the optimized D2FC models had high error scores across evaluation criteria (Fig. 4b), failing to recapitulate the system's emergent property (Supplementary Fig. 5c and f). Since the D2FC model failed to recapitulate the data despite a broad exploration in parameter space, we conclude that its architecture lacks critical regulatory components necessary to explain the experimentally observed live-cell dynamics.

Pseudo-zero-order processes are unusual behaviors in biological systems because they appear to be independent of reactant concentrations. Typically, zero-order kinetics result from systems where one of the reactants is greatly limited or scaffolded into a lower reactivity state. Since NF-κB is well established as a transcription factor, we hypothesized that NF-κB binding broadly to DNA[45–49] may be an essential mechanism that is not explicit in the D2FC model. Furthermore, there is a growing body of literature that characterizes the chromatin remodeling capabilities of NF-κB. For example, NF-κB can bind to assembled nucleosomes and flanking DNA, contributing to DNA unwrapping and nucleosome displacement[50,51]. Chromatin reorganization was found to decode the duration of nuclear NF-κB, where prolonged nuclear TF contributes to stimulus- and gene-specific transcription initiation by progressively revealing promoter sequences that are otherwise closed in basal conditions[52]. More recently, prolonged non-oscillatory nuclear NF-κB was shown to cause cytokine-inducible alterations to chromatin accessibility near NF-κB binding sequences, activation of latent enhancers, and epigenetic reprogramming[23]. Therefore, additional mechanisms were added to

the model to include NF-κB binding to DNA and chromatin remodeling that reveals additional NF-κB binding sites via mechanisms akin to pioneering[53] (see methods). The modified D2FC model with chromatin remodeling is referred to as D2FC-squared (D2FC²) henceforth (see methods, Supplementary Tables 1 and 2).

Using PSO, we again sampled 500 parameter sets with the D2FC² architecture. The D2FC² model produced high quality fits, significantly reducing scores of the 3 error criteria (Fig. 4b). Parameterization sets of the D2FC² from PSO showed that the approach effectively explored the prior distribution of acceptable parameters, and the top 10 scoring models were able to achieve a similar quality of results despite being derived from distinct combinations of parameters (Fig. 4b and Supplementary Fig. 6). Taken together, the topology of D2FC² enables the model to recapitulate the average behaviors of all experimental conditions, with robustness to variation of parameters. The D2FC² model indicates that the 4×1.5-minute pulse promotes a more permissive chromatin state and increased total NF-κB DNA binding, resulting in a slower relative accumulation of NF-κB-regulated gene products and reduced rates of nuclear export. For points of comparison, we also tested two alternative D2FC architectures (Supplementary Fig. 7): one incorporating IκBε, an NF-κB–inducible gene with a 45-minute delay relative to IκBα production[54], and the other incorporating IκBβ, which is constitutively expressed and not inducible by NF-κB[43,55] (see methods, Supplementary Tables 5 and 6). Both model variants were optimized using particle swarm optimization, and neither recapitulated the emergent property, nor outperformed the D2FC² model in other error score metrics (Supplementary Fig. 7).

## A model with chromatin reorganization accurately predicts single cell responses and experiments

Training and validation simulations with the D2FC² model used averages of experimental single cell time courses for each condition, both as inputs to simulations and to evaluate the quality of fits. We next asked whether the D2FC² accurately predicts single-cell dynamics. Using the 3 error criteria to rank the models, the top-performing parameter set in the D2FC² was selected for further analysis (Fig. 4b, red asterisk). Single-cell EGFP-NEMO time-courses from all the single pulse and pulse train conditions were used as inputs to simulate nuclear NF-κB responses. Visually, simulated single-cell time courses of nuclear NF-κB appeared like experiments, showing both first-order and zero-order nuclear export kinetics appropriate to each condition (Fig. 5a and Supplementary Fig. 5g). AUCs of simulated nuclear NF-κB to single pulse conditions showed dose-responsiveness, and consistent with expectations based on error scores following PSO, D2FC² simulations for pulse trains recapitulated the emergent property (Fig. 5b and c). Furthermore, interquartile ranges of boxplots were similar when comparing experiments and simulations, suggesting that cell-to-cell variability at the level of CI dynamics contributes significantly to the variability of nuclear NF-κB dynamics observed in single-cell experiments.

Following, we moved away from population average data and evaluated the quality for single cell fits by measuring the SSE between experimental nuclear RelA trajectories and simulated NF-κB dynamics. An elbow plot was used to define the error threshold to classify low- and high-quality fits, and a stricter threshold was applied to define a third class of 'excellent fits' where experiment and simulations overlap almost perfectly (Fig. 5d and Supplementary Fig. 8a). When compared between experimental conditions, performance of the D2FC² was consistent with most single cells showing high- and excellent-quality fits (Supplementary Fig. 8b). The single 15-minute pulse and the 3x2min were the conditions that showed the most low-quality fits; nevertheless, even these weakest conditions still achieved excellent or high-quality fits for approximately 70% of single cell trajectories. Comparing the top 10 D2FC² parameter sets, all recapitulate single pulse dose-responsiveness and the emergent property, as well as

produce high-to-excellent predictions for over 80% of the single cell nuclear RelA trajectories aggregated across all conditions (Fig. 5e; see also Supplementary Table 3). These results demonstrate that many different parameterizations of the D2FC² architecture can robustly resolve single-cell EGFP-NEMO time courses and predict nuclear RelA responses across a heterogeneous population of cells.

Finally, we sought to validate the core assumptions of the D2FC² model architecture by experimentally testing model predictions that single and multi-pulse stimuli differentially regulate the extent of NF-κB binding to DNA and the pioneering of accessible chromatin. The model predicts that NF-κB–DNA binding will be increased following a 4×1.5-minute pulse when compared to a single 6-minute pulse at both 60 and 180 minutes after stimulation (Fig. 6a). Previously, experiments have measured immobilization of FP-tagged TFs on DNA in living cells using Fluorescence Recovery After Photobleaching (FRAP)[56]. We therefore used FRAP to measure nuclear NF-κB diffusion following photobleaching (Fig. 6b) and calculated the amount of immobilized NF-κB bound to DNA from the fluorescence recovery curves (Fig. 6c and Supplementary Fig. 9a). Consistent with D2FC² model predictions at both timepoints, stimulation with a 4 × 1.5-minute pulse results in a significant increase of nuclear NF-κB immobilized on DNA.

Although the D2FC² only simulates NF-κB, IL-1–stimulation regulates other DNA-binding complexes in parallel that influence chromatin structure with similar temporal dynamics, such as AP-1[57]. We therefore considered the cumulative AUC of NF-κB bound to DNA as a surrogate in proportion with the broader IL-1-induced pioneering effects that are not explicitly modeled. Comparison of the 4×1.5 and 1×6-minute pulse patterns predict similar levels of chromatin accessibility at 60 minutes, but a substantial increase in the 4×1.5-minute condition by 180 minutes (Fig. 7a). To test this model prediction, we used immunofluorescence (IF) to measure levels of the histone modification H3K4me3, a marker of open and transcriptionally active chromatin, based on prior experiments that show enhanced H3K4me3 staining intensity following cytokine stimulation[58]. Although NF-κB is not a histone methyltransferase, NF-κB–target genes have been shown to exhibit increased H3K4me3 through direct recruitment of MKL1, which facilitates additional recruitment of histone methyltransferases (HMT) to active promoters[59]. Similarly, NF-κB interacts with the MLL1 HMT complex both in the cytoplasm and the nucleus, and MLL1 disruption has been implicated with reduced NF-κB target gene expression without disrupting NF-κB activation nor its DNA-binding profile[60]. In our microfluidic system, we quantified H3K4me3 IF staining (Fig. 7b) in cells exposed to either the 4 × 1.5 or 1 × 6-minute pulse pattern. Consistent with model predictions, differences in H3K4me3 levels between the two conditions were modest at 60 minutes and enhanced significantly in the multi-pulse condition at 180 minutes (Fig. 7c and Supplementary Fig. 9b). OICR-9429 is a small-molecule inhibitor that disrupts MLL1 interactions and reduces MLL1-specific H3K4me3 levels in treated cells[61]. We therefore pretreated cells with 10 µM OICR-9429 for 24 hours, followed by stimulation with a 4×1.5-minute pulse train and observed that H3K4me3 levels remained baseline at 180 minutes (Fig. 7c). Despite a loss of H3K4me3, and consistent with previous observations[60], OICR-9429 had minimal impact on the DNA-immobilized fraction of NF-κB following stimulation (Fig. S9c). These findings suggest H3K4me3 methylation is a secondary event that follows nuclear NF-κB interactions and other pioneering events, supporting a mechanistic role for MLL1-dependent H3K4me3 following a cytokine pulse train. Taken together, FRAP and IF results provide experimental support for the two core model mechanisms and associated predictions, that multi-pulse stimulation enhances NF-κB binding to DNA and permissiveness of chromatin.

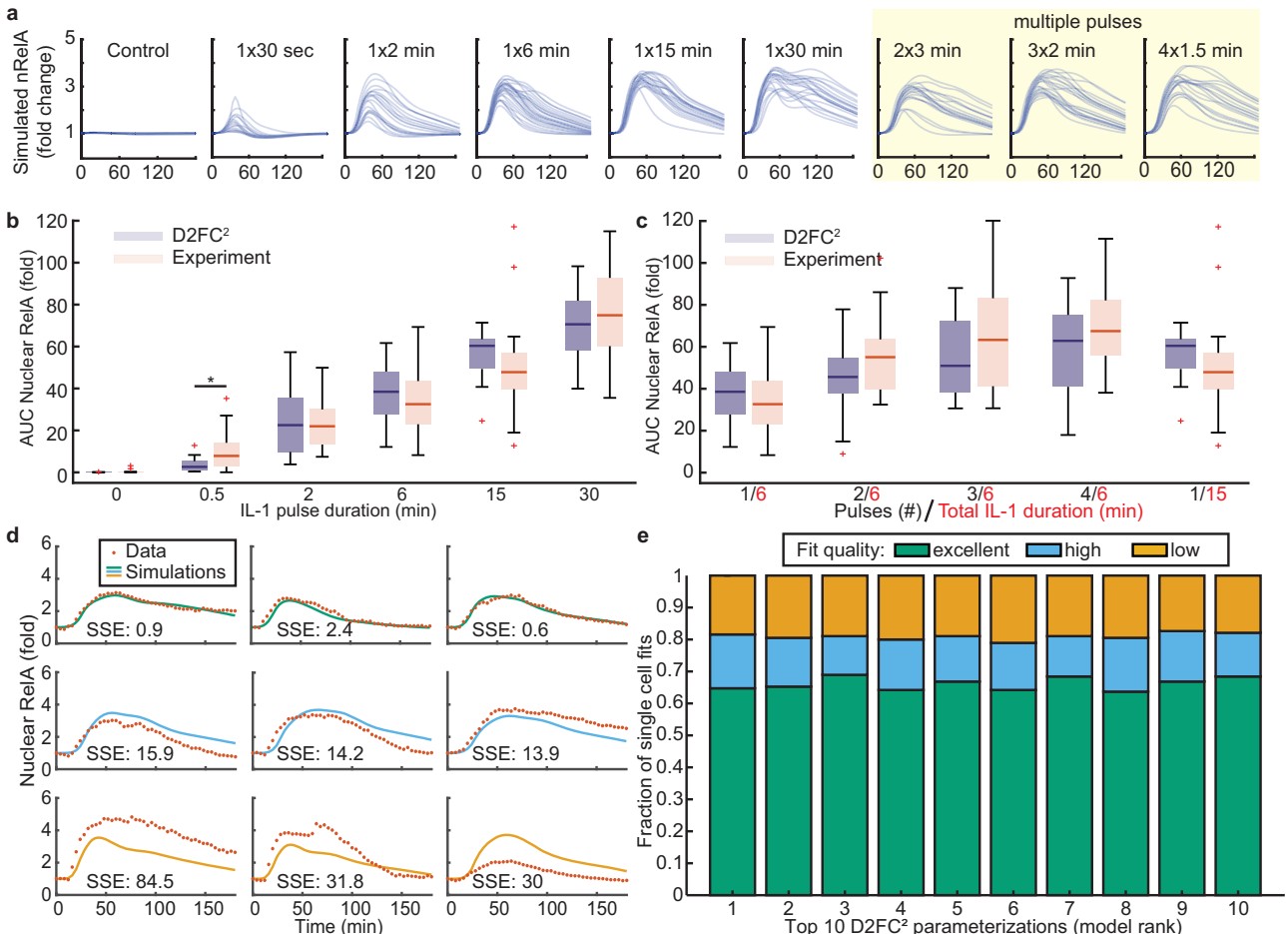

**Fig. 5 | D2FC² accurately predicts NF-κB responses using single cell CI data as inputs. a** Simulated single-cell NF-κB predictions using the best-performing D2FC² model using single-cell EGFP-NEMO as model inputs. **b, c** Boxplots comparing the simulated and experimental AUC of nuclear RelA for single-pulse (**b**) and multi-pulse (**c**) dosing schedules. Between experiments and D2FC² simulations, the model shows remarkable similarity where only one condition has a statistically distinct difference (marked with *; p = 0.01, student's two-sided t-test). For single pulse data (**b**), n values are: 0 min = 20; 0.5 min = 18; 2 min = 20; 6 min = 28; 15 min = 22; 30 min = 21. For multi-pulse data (**c**), n values are: 1/6 = 28; 2/6 = 23; 3/6 = 19; 4/6 = 19; 1/15 = 22. **d** Example simulated single cell trajectories highlighting the excellent, high,

and poor single cell trajectories compared to the experimental data. **e.** Proportion of single-cell predictions classified as excellent, high, or low quality (see also, Supplementary Fig. 7) compared to experimental data of the top 10 models in predicting single-cell trajectories aggregated across all experimental conditions. See also Supplementary Table 3. For boxplots, the boxes represent the interquartile range with the median indicated as the center line. The bounds of the box correspond to the 25th and 75th percentiles. Whiskers extend to the minima and maxima values within 1.5 interquartile range, and points beyond are shown as outliers. Source data are provided as a Source Data file.

## Peak number and adaptation time of CI fine tunes chromatin permissiveness and nuclear NF-κB dynamics

Temporal profiles of IKK activity encode information about the abundance and type of inflammatory factors in the milieu, regulating nuclear NF-κB and stimulus-specific gene expression programs[17,55]. Our results here show that features of IKK at CI assemblies also encode information about the temporal presentation of cytokines, such as their numbers and timing, that in succession regulate feedback pathways in the nucleus (Fig. 8a).

Based on single-cell time courses of EGFP-NEMO from static cytokine concentrations[17], single cytokine pulses (Fig. 1), and cytokine pulse trains (Fig. 2), CI-encoding can be decomposed into a primary peak and a second slow-adapting distribution (Fig. 8b). We leveraged the D2FC² to systematically evaluate the impact of CI-encoding features on downstream signaling (Fig. 8c). First, we considered nuclear NF-κB fold change, which we showed previously to correlate strongly with NF-κB-regulated early response genes[39]. Here, maximal nuclear NF-κB correlates with peak CI numbers that saturate around 300, and adaptation time for the secondary distribution had negligible impact (Fig. 8c, top left). Comparing rates of nuclear NF-κB export showed a

non-linear relationship with CI-encoding features. A primary peak of approximately 50 CI puncta and adaptation times of greater than 80 minutes were both strictly required for conversion of nuclear NF-κB export rates from a first-order to pseudo-zero-order process (Fig. 8c, top right). Higher peak CI numbers required even longer adaptation times to achieve zero-order NF-κB export kinetics, possibly requiring greater CI persistence to compensate for stronger first-wave transcription negative feedback mediators, such as IkBα. Chromatin openness showed a similar pattern, requiring a minimum peak CI of nearly 100 puncta, and prolonged activation to increase overall permissiveness (Fig. 8c, bottom left). Even the highest peak CI numbers failed to enhance chromatin opening unless adaptation time also persisted for more than 60-80 minutes. Finally, since open chromatin reveals new NF-κB binding sites, both productive and non-productive in regulating gene transcription, we measured the fraction of free nuclear NF-κB as a function of chromatin status. Remarkably, even when concentrations of nuclear NF-κB are at their highest, permissive chromatin can deplete free nuclear NF-κB (Fig. 8c, bottom right). Therefore, despite high total concentrations of nuclear NF-κB, permissive chromatin at later times may only leave trace amounts of free

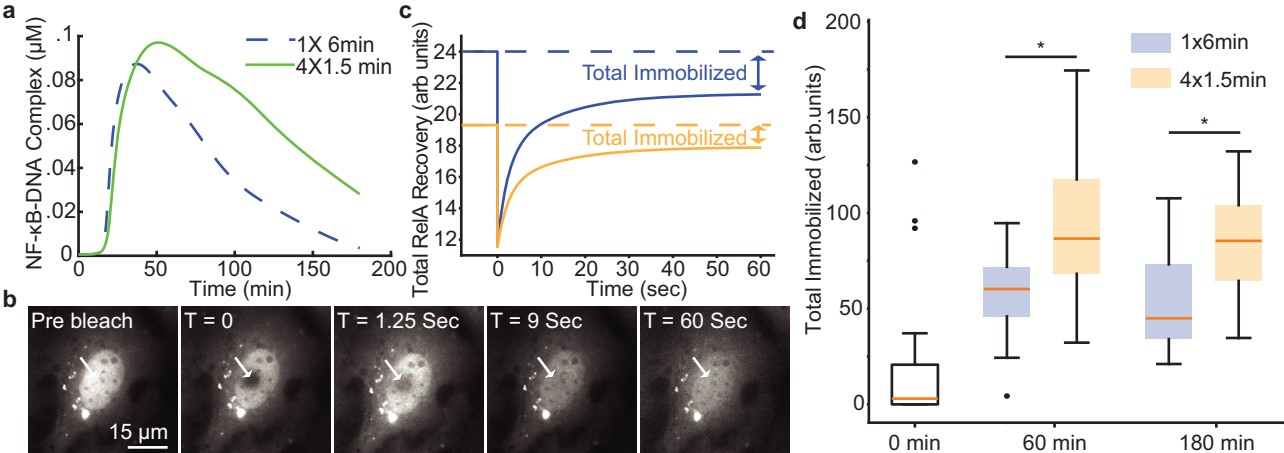

**Fig. 6 | A pulse train enhances nuclear NF-κB immobilization. a** Model predictions of NF-kB-DNA binding were generated by inputting the average IKK spot trajectories from the 1×6-minute and 4×1.5-minute pulse profiles into the optimized D2FC² model. **b** Example images of the FRAP experiment highlight the recovery curve's key points. For each condition, FRAP recovery curves were collected from 20 independent cell nuclei. **c** Explanatory schematic of a RelA recovery trajectories illustrating how the total immobile fraction is estimated from fluorescence recovery curves, see also Fig. S9a. **d** Boxplot summarizing changes in the total immobile fraction across different pulse patterns. Student's two-sided t-test p-values for pairwise comparisons; 1 × 6 vs. 4 × 1.5 (60 min) = 0.001; 1 × 6 vs. 4 × 1.5 (180 min) = 0.002. Student's t-tests revealed significant differences (p < 0.01, student's two-sided t-test) for all comparisons, except between the 60- and 180-minute time-points within each pulse pattern, where no statistical significance was observed. For each condition, 20 cells were analyzed. For boxplots, the boxes represent the interquartile range with the median indicated as the center line. The bounds of the box correspond to the 25th and 75th percentiles. Whiskers extend to the minima and maxima values within 1.5 interquartile range, and points beyond are shown as outliers. Source data are provided as a Source Data file.

nuclear TF to interact with promoters of early response genes such as IkBα.

Studying I/O relationships in dual-reporter cells exposed to dynamic stimuli, while leveraging naturally occurring cell-to-cell variability, revealed features of CI encoding that regulate time profiles of nuclear NF-κB. Simulations with the D2FC² show that the primary peak of CI determines the fold change of nuclear NF-κB and the adaptation time facilitates switching towards more permissive chromatin states with additional NF-κB binding sites. Importantly, DNA-based sequestration of nuclear NF-κB is transient, and the model allows nascent IkBα polypeptides to interact with DNA-bound NF-κB to facilitate dissociation, as seen in experiments[62]. Together, our experiments and simulations suggest that the transition to a pseudo-zero-order export process results from the combined effect of sequestration and feedback depletion on early gene promoters.

## Discussion

On the surface of a human cell, the number of IL-1 and TNF receptors is relatively small, estimated in the range of hundreds to low thousands per cell[17,63]. Following cytokine stimulation, ligated receptors are rapidly internalized and degraded[64,65]. Therefore, a limited and depletable resource of surface receptors is available to encode information about the inflammatory milieu into the cell via CI assemblies. We reasoned that cytokine conditions that spread out a stimulus in time may encode different signals from tonic and saturating boluses that would rapidly deplete the available receptor pool. To this end, our result shows that temporal properties of IKK encode information about time-varying extracellular conditions that can be used by the cell in succession to regulate distinct temporal response patterns of nuclear NF-κB.

We used dynamic stimuli, dual-reporter cells, and computational models to investigate information encoding properties of the IKK-NF-κB signaling axis. Exposing dual-reporter cells to a single cytokine pulse revealed that the aggregate AUCs of IKK and nuclear NF-κB increase with the IL-1 pulse duration, forming a continuum of single-cell I/O responses (Fig. 1f). In contrast, when a cytokine is presented as a series of short pulses, monotonicity of the aggregate I/O response is disrupted with cells showing disproportionately enhanced AUCs for

nuclear NF-κB responses (Figs. 2 and 3). We determined that the aggregate AUC of IKK puncta is not sufficient to predict same-cell NF-κB responses to dynamically presented stimuli. Instead, orthogonal axes for peak numbers and adaptation times of CI were predicted to encode signals that enhance NF-κB responses by mechanisms that switch nuclear export from a first-to pseudo-zero-order process. Although both features are important contributors, CI adaptation times where EGFP-NEMO puncta persist for around 80 minutes or longer define a threshold for conversion to zero-order kinetics. This observation is reminiscent of a previous computational prediction where a constant plateau of low-amplitude IKK activity can mediate long-lasting nuclear NF-κB time profiles[55]. Nevertheless, experiments differ from these predictions because CI assemblies and IKK kinase assays in wild-type cells[16,17,55,66] show adaptation, typically within 60-90 minutes following exposure to TNF or IL-1. Based on our experiments and model results, we surmise that the tail of IKK activity that extends beyond 80 minutes is crucial to blunt the first wave of IkBα-mediated feedback, subsequently facilitating nuclear remodeling that supports zero-order export kinetics.

In contrast with experiments, and regardless of model parameterization, simulations with the D2FC model[39] to predict NF-κB responses did not show switching between first- and zero-order nuclear export kinetics, and the emergent property with enhanced response AUCs. The D2FC² modified the base model by invoking explicit DNA-binding and chromatin-remodeling mechanisms for NF-κB. These were selected because they reflect increasingly well-characterized mechanisms that can create reactant-limiting conditions necessary for a pseudo-zero-order process. To illustrate this point, NF-κB binds to hundreds of non-redundant sequences that are distributed with repetition throughout the human genome[47,67]. Early estimates predicted that there are significantly more binding sites in the genome than the ~$10^5$ NF-κB molecules in a typical mammalian cell[45]. Subsequently, tens of thousands of NF-κB binding sites have been observed in macrophages and B cells, as well as enrichment of NF-κB binding on the 5' end of genes or other non-promoter sequences[46,48,49]. More recent results using genetic knockout of IkBα in mouse BMDMs showed prolonged nuclear NF-κB following TNF stimulation, along with disruption of nucleosomal histone-DNA

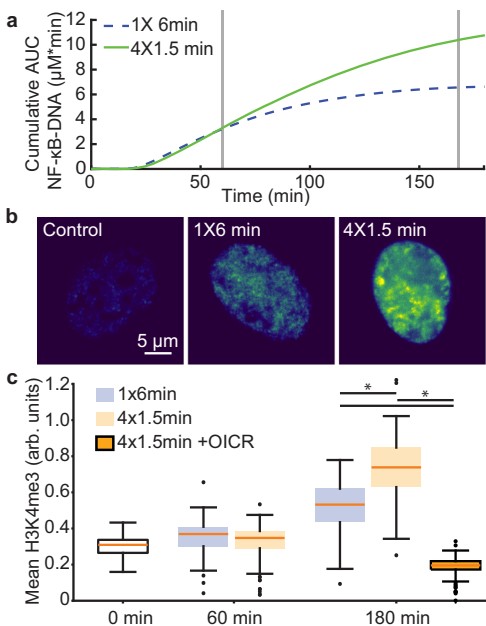

**Fig. 7 | Pulse trains enhances nuclear H3K4me3 hours after stimulation.**
**a** Model predictions of cumulative area under the curve (AUC) NF-κB -DNA binding, which served as a model estimate for chromatin re-modeling, were generated by inputting the average IKK spot trajectories from the 1×6-minute and 4×1.5-minute pulse profiles into the optimized D2FC$^2$ model. **b** Representative images of nuclear H3K4me3 immunofluorescence staining following treatment in the microfluidic chip with control, 1X6-minute, or 4X1.5-minute pulse profiles, acquired 180 minutes post-stimulation. **c** Boxplot summarizing H3K4me3 immunofluorescence intensity across conditions, including OICR-9429 treated cells. Values for n are: control = 110; 1×60 (60 min) = 93; 4×1.5 (60 min) = 121; 1×6 (180) min = 72; 4×1.5 (180 min) = 80; OICR 4x1.5 (180 min) = 128. Student's two-sided t-test p-values for pairwise comparisons at 180 min are: 1×6 vs. 4×1.5 = 10$^{-12}$; 1x6 vs. OICR-4×1.5 = 10$^{-53}$; 4×1.5 vs. OICR-4×1.5 = 10$^{-76}$. For boxplots, the boxes represent the interquartile range with the median indicated as the center line. The bounds of the box correspond to the 25th and 75th percentiles. Whiskers extend to the minima and maxima values within 1.5 interquartile range, and points beyond are shown as outliers. Source data are provided as a Source Data file.

interactions in the vicinity of NF-κB binding sites[23]. Since earlier studies did not use genetic or chemical perturbations that are necessary to produce prolonged nuclear NF-κB responses and chromatin reorganization in differentiated immune cells, they are likely to underestimate the breadth of NF-κB-DNA interactions. Taken together, there is a vast abundance of productive and non-productive NF-κB-DNA interactions that are basally accessible, and significantly more following chromatin reprogramming[52]. Incorporation of these roles in the D2FC$^2$ revealed that the consequences of DNA-binding and chromatin are sufficient to switch nuclear NF-κB export to a pseudo-zero-order process. Consequently, our results also suggest that mechanisms associated with chromatin dynamics and epigenetic reprogramming may be selected via dynamic stimuli under flow that control the timing and numbers of CI. There is growing recognition of multi-hit immune signaling where cytotoxicity is additive over multiple sub-lethal interactions between immune and cancer cells[68,69]. With these observations, our results support that dynamic cytokine presentations, such as multi-hit pulses, may encode distinct messages to receiving sub-populations of non-immune cells, including cancers.

By using PSO to fit averaged experimental data from four conditions that embody the emergent property, families of model parameterizations were identified that successfully recapitulate experiments. As discussed earlier, the D2FC$^2$ model architecture was modified for chromatin permissiveness and DNA binding only, and did not invoke any additional molecular species or mechanisms. Simulations in silico

further demonstrated capabilities of D2FC$^2$ to accurately predict single cell NF-κB responses from time courses of CI, validated with a compendium of experimental trajectories from dual reporter cells. Given that there were no single-cell data used during optimization, it was remarkable that a family of different model parameterizations was found, each capable of accurately predicting same-cell NF-κB responses from experimental EGFP-NEMO time profiles. The added mechanisms of D2FC$^2$ are robust, experimentally supported, and allow the resulting system to achieve consistent behavior without requiring strict expression levels for all molecular species in the model that individually tend to exhibit variability when measured in single cells. Taken together, low numbers for cytokine receptors and stochasticity of their interactions are among the chief contributors to heterogeneity between single-cell NF-κB responses, which can be accurately predicted from CI-level measurements.

Using experimental CI trajectories as input to simulations and evaluating model outputs against experimental data were critical for model selection. A current conceptual gap in the model is the ability to simulate CI and IKK responses to arbitrarily complex environments and similarly, identify which theoretical distributions can be achieved experimentally. While explicit models of CI are feasible, an important caveat is that IKK has been shown to form ubiquitin-induced liquid-liquid phase separated droplets at high concentrations[70,71]. Although droplets versus protein complexes do not alter the interpretation of our data, future models may require biophysical considerations for condensates, or even a simpler model as previous developed where each CI was modeled independently and fit to fluorescence intensity curves[17]. Nevertheless, we expect that forthcoming models will explicitly model the extracellular milieu, CI puncta, NF-κB dynamics, and gene-specific promoters relative to chromatin. Through a comprehensive model, we will understand how basal conditions of different cell types encode and decode certain cytokine responses, and how complex environments can select gene expression programs using CI and IKK as signaling hubs.

Dynamics of nuclear NF-κB mount stimulus-specific adaptive responses through selective regulation of gene expression programs, many of which cluster into early-, mid-, and late-response categories[22,55,72]. Although late response genes are typically associated with tonic and high-concentration inflammatory stimuli, this may not be strictly required. Based on our experiments and model, we speculate that mid- and late-response genes can arise from chromatin accessibility, and that certain temporal stimuli that distribute CI numbers in time have the potential to reveal their promoters via chromatin remodeling. Our results, therefore, suggest that distinct classes of NF-κB response states, predicted from information theory analyses and experiments, may be achievable via dynamic extracellular stimuli or other conditions that precisely manipulate CI. Future works should explore the impact CI dynamics on classes of NF-κB responses, coupled with differential gene expression and chromatin-mediated mechanisms, to further define these states. Consistent with these principles, recent computational analysis of single-cell responses to dynamic IFN-γ stimuli identified stochastic regulation of chromatin-accessibility as a mechanism that enhances cell-to-cell heterogeneity[73]. Additionally, the mechanism for chromatin remodeling in the D2FC$^2$ is likely to be an oversimplification, lacking mechanisms for gene-specific promoter accessibility and other DNA-binding proteins such as AP-1 that remodel chromatin in response to cytokines[74,75]. We anticipate these mechanisms will be important factors in future models that refine our understanding of cytokine responses, cell states, and contributors to single-cell heterogeneity.

In summary, our results demonstrate that the number and timing of CI assemblies encode information about time-varying stimuli in the extracellular milieu. With limited numbers of surface receptors to nucleate CI assemblies, extracellular conditions that prolong the adaptation time of CI disproportionately enhance the aggregate NF-κB

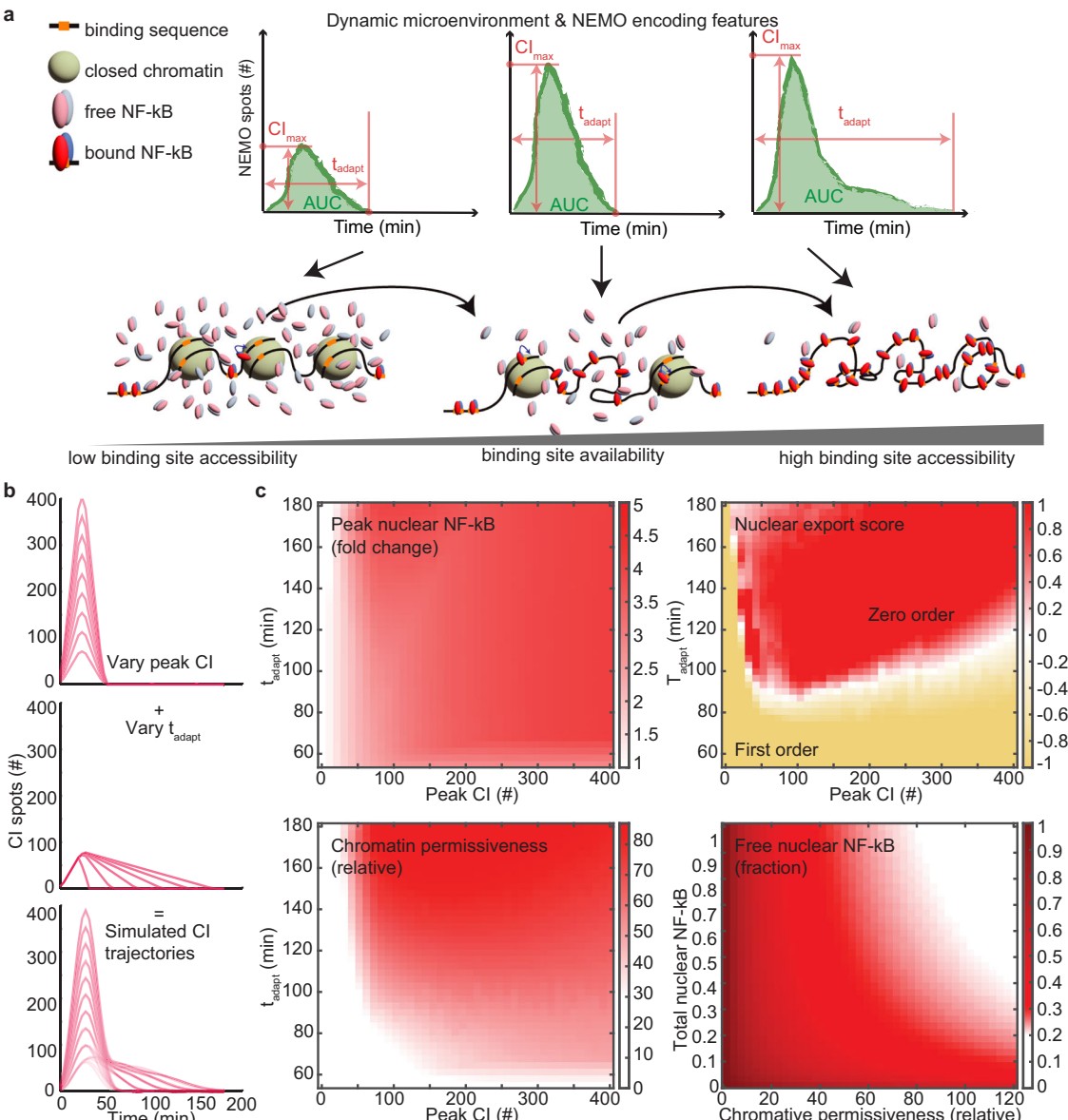

**Fig. 8 | Features of NF-κB responses show different sensitivities to features of CI. a** Schematic illustrating how NEMO features encoded by temporal stimulation impacts binding site availability and the duration of NF-κB retention. **b** Simulated IKK trajectories varying the peak of NEMO numbers (top, peak), adaptation time of NEMO numbers (middle, $t_{adapt}$) independently. Combined features are used to generate simulated time-courses that scan the encoding space of NEMO puncta (bottom). **c** Heatmap of: simulated peak nuclear RelA (fold change) with varying peak NEMO and adaptation times (top-left); Nuclear RelA export scores for simulated responses, where positive values indicate increasingly zero-order scores and negative values indicate the strength of first-order scores (top-right); Relative chromatin permissiveness, where higher values represent more open chromatin (bottom-left); and results from a sub-model used to simulate the steady-state fraction of free nuclear NF-κB changes with overall nuclear NF-κB and chromatin permissiveness (bottom-right).

response. These observations are recapitulated in a model of NF-κB signaling by invoking mechanisms for DNA-binding and chromatin reorganization. The resulting system reveals how IKK-encoding of dynamic environmental conditions distinctly coordinates a range of potential NF-κB responses through systems feedback and chromatin remodeling.

## Methods
### Cell culture
U2OS cells (ATCC) were previously CRISPR-modified to express N-terminal fusions of EGFP-NEMO and mCherry-RelA from their endogenous loci[17,76], as well as KYM-1 cells stably expressing mVen-RelA[29], were cultured in McCoy's 5 A and RPMI medium, respectively, at 37 °C in a humidified incubator with 5% $CO_2$. The medium was supplemented with 10% fetal bovine serum (FBS; Corning, 35-010-CV,

Lot: 03322001), penicillin/streptomycin (100 U/mL; Corning, 30-002-CI), and 0.2 mM L-glutamine (Corning, 25-005-CI). Cells were routinely screened for mycoplasma contamination.

### Live-cell imaging for pulse experiments in the microfluidic dynamical stimulation system
A custom microfluidic system, as previously described[30], was used for single and repeat pulse experiments. Briefly, two-inlet PDMS (1:10 ratio of hardening agent to PDMS base; DOW, Sylgard 184) devices were fabricated from corresponding 3D-printed molds, sterilized by auto-claving, washed with ethanol, followed by PBS, and subsequently incubated with a 0.002% (v/v) fibronectin solution (Sigma-Aldrich, F1141) in PBS for 24 hours at 37 °C. U2OS double-CRISPR cells (~ 5 × 10⁶ cells/mL) were seeded into the microfluidic devices and incubated for at least 24 hours. For experiments involving OICR-9429

treatment (MedChemExpress, HY-16993), cells were seeded and cultured in the microfluidic chip for 24 hours prior to drug treatment. They were then incubated with 10 µM OICR-9429 for an additional 24 hours before the pulse experiment.

On the day of the experiment, the microfluidic device was connected using Tygon tubing to fluid reservoirs on the gravity pump containing the appropriate treatments. FluoroBrite DMEM medium (Gibco, A18967-01) supplemented with 10% FBS (Corning), penicillin (100 U/mL), streptomycin (100 U/mL), and 0.2 mM L-glutamine was used during imaging. IL-1 and TNF treatments (Peprotech, 200-01B and 300-01 A, respectively) were prepared at indicated concentrations with Alexa Fluor 647–conjugated BSA (0.0025% v/v of a 5 mg/mL stock; Invitrogen, A34785) to visualize the cytokine-containing stream in the device.

Images of EGFP-NEMO were acquired every 4 minutes using FITC filters, with a z-stack of eight images at 0.5-µm intervals, using 0.04-second exposure and 32% transmission. Similarly, mCherry-RelA was imaged every 4 minutes using the Alexa A594 filters, with 0.1-second exposure and 50% transmission. Alexa Fluor 647–conjugated BSA was imaged every 1 minute using the CY5 channel with 0.1-second exposure and 50% transmission. All images were acquired in an environmentally controlled chamber (37 °C, 5% $CO_2$) on a DeltaVision Elite microscope equipped with a pco.edge sCMOS camera and an Insight solid-state illumination module (GE Healthcare) at 60x LUCPLFLN oil objective.

## Fixed-cell immunofluorescence

U2OS double-CRISPR cells were seeded into microfluidic chips and stimulated with IL-1 (10 ng/mL) using either a single 6-minute pulse (1×6) or four 1.5-minute pulses (4×1.5). Cells were imaged and fixed directly within the microfluidic chips at either 60 or 180 minutes. After removing the Tygon tubing from the inlets and outlets, unfiltered pipette tips were inserted to allow manual exchange of solutions. The cell culture chamber was washed with PBS, then cells were fixed with 4% paraformaldehyde and permeabilized with 100% methanol, each for 10 minutes at room temperature. Between each step, 400 µL of PBS was washed through the chamber. Cells were blocked with 2% BSA (Fisher Scientific, BP9700100) in PBS overnight at 4 °C. Cells were then incubated with anti-H3K4me3 antibody (1:100 dilution; Thermo Fisher Scientific, 711958; Lot: 2977681) in 2% BSA, either overnight at 4 °C or for 1 hour at room temperature. After washing with 400 µL of 2% BSA in PBS, cells were incubated for 1 hour at room temperature with a secondary antibody solution containing 1:2000 Alexa Fluor 647 donkey anti-rabbit (Thermo Fisher Scientific; A-31573) and 200 ng/mL Hoechst in 2% BSA in PBS. Finally, secondary antibody solution was replaced with PBS before being imaged at 60X magnification on an Olympus IX83 microscope equipped with a Cicero widefield/confocal system. Mean fluorescence intensity of H3K4me3 staining was quantified by segmenting nuclei using CellPose 2.0[77] on the Hoechst channel and measuring the signal within each region using custom Python scripts.

## Fluorescence recovery after photobleaching (FRAP)

FRAP was performed on an Olympus IX83 microscope equipped with a UGA-42 photomanipulation system (Rapp OptoElectronics), a 60X oil immersion objective controlled by CellSens software. U2OS double-CRISPR cells expressing endogenous mCherry-RelA were maintained at 37 °C in a live-cell imaging chamber and stimulated with IL-1 (10 ng/mL) using either a 1×6-minute or 4×1.5-minute pulse pattern. FRAP imaging was conducted within 15 minutes of either the 60- or 180-minute post-stimulation time point. Imaging began with acquisition of 10 pre-bleach frames of mCherry-RelA, followed by photobleaching of a 5 µm radius region in the nucleus using a 405 nm laser at 100% power for 2 seconds. Fluorescent recovery was imaged with 0.25-second intervals for 1 minute using 555 nm excitation with 100 ms exposure for each frame. To estimate the fluorescent recovery curve, the data were processed using the double normalization method as described

previously[78]. Recovery curves were then fit to a two-component fit as previously reported[79]:

$$F(t) = y_0 + A_1\left(1 - e^{-\frac{t}{\tau_1}}\right) + A_2\left(1 - e^{-\frac{t}{\tau_2}}\right) \quad (1)$$

Where F is the percent recovery, t is the time, and $y_0, A_1, A_2, \tau_1,$ and $\tau_2$ are parameter found through optimization. The total immobilized was determined by multiplying the recovery curve by the nuclear mCherry-Rela fluorescent intensity on the first frame before bleaching.

## Extracting EGFP-NEMO and mCherry-RelA time profiles from live-cell images

As previously decribed[80,81], dNEMO software was used to detect and quantify EGFP-NEMO spots. The spot detection threshold in dNEMO was set between 2.1 and 2.4, and only spots present in at least two contiguous slices of the 3D images were considered valid. Individual cells were manually segmented using dNEMO's keyframing function. The mCherry-RelA nuclear intensity trajectories were quantified using custom python scripts. Briefly, the custom scripts allowed for identifying regions of background and the nucleus across all frames by defining regions by a box. Nuclear NF-κB was determined by calculating the mean pixel intensity of the nucleus by the mean background intensity of the frame. Each cell trajectory was manually tracked across all images. To calculate the fold change, mCherry-RelA trajectories were divided by the initial nuclear fluorescence at time zero. The area under the fold-change curve was calculated by first subtracting by 1 and setting any negative values equal to 0. The trapezoidal function in MATLAB was used to calculate the area under the curve from the data.

$$NFkBFC\_AUC = trapz(max(NFkBFC - 1, 0)) \quad (2)$$

Where NFKBFC represents the fold change nuclear NF-κB.

## Mechanistic modeling

The mechanistic model of NF-κB activation through IKK, called D2FC, was used as previously published[39]. Briefly, the mechanistic model contains the key events from IKK activation, degradation of IkB due to IKK, NF-κB translocation to the nucleus, and the up-regulation of IkB in the cell. To interface the single-cell measurements of IKK punctate formation, the D2FC model was modified such that the activation of IKK from its neutral state depended on the formation of IKK punctate structures. The rate of formation is:

$$IKK_n \rightarrow IKK \quad (3)$$

$$Rate = k_a * IKKSpots(t) * IKK_n \quad (4)$$

The parameter $k_a$ was optimized to reduce the sum of the squared error of the training data and represents the parametrization of D2FC.

The input, IKKSpots(t), consists of experimental trajectories obtained from live-cell imaging. These trajectories were fitted to a sum of four Gaussians using MATLAB's fit function. To ensure that the fitted curve started at zero, additional weight was placed on the initial value during optimization. Finally, all simulated IKK trajectories were visually inspected to confirm that they captured the basic trends observed in the experimental data.

$$IKKSpots(t) = \sum_{i=1}^{4} a_i e^{\left[-\left(\frac{(t-b_i)}{c_i}\right)^2\right]} \quad (5)$$

The parameters for each single cell IKKSpot trajectory and mean IKKSpot trajectory can be found in the ModeledIKKTrajectories.xlsx supplementary information.

The D2FC$^2$ model introduced an additional mechanism that was necessary to fit the emergent property as described in the results. The new mechanisms involved NF-κB binding and opening new DNA-binding sites in chromatin. This mechanism includes a new model species, NFκBDNA, representing NF-κB bound to DNA. The binding rate of NF-κB to DNA is influenced by cooperative effects, where the presence of already-bound NF-κB increases the likelihood of additional NF-κB binding. This phenomenon reflects a so-called pioneering effect[53] in which the binding of NF-κB facilitates chromatin relaxation, opening more binding sites and enhancing NF-κB recruitment, where $Ps_0$ represents the basal nuclear openness and was fixed to a value of 1:

$$NFkB \rightarrow NFkBDNA \tag{6}$$

$$Rate = ka1d * DCoop * NPio \tag{7}$$

$$DCoop = \frac{\left(\frac{NFkB}{kdNFKB}\right)^{h2}}{1 + \left(\frac{NFkB}{kdNFKB}\right)^{h2}} \tag{8}$$

$$NPio = Ps_0 + Ps * \frac{\left(\frac{NFkBDNA}{KDNA}\right)^{h3}}{1 + \left(\frac{NFkBDNA}{KDNA}\right)^{h3}} \tag{9}$$

NF-κB can dissociate from DNA either through direct binding of nuclear IκBα or via a basal rate of dissociation. The two methods follow basic mass action kinetics

$$IkBa + NFkBDNA \rightarrow IkBaNFkB \tag{10}$$

$$Rate = ka2a * [N.IkBa] * [N.NFkBDNA] \tag{11}$$

$$NFkBDNA \rightarrow NFkB \tag{12}$$

$$Rate = kd1d * [NFkBDNA] \tag{13}$$

**Emergent property score**

The emergent property score was calculated by first simulating nuclear RelA trajectories for each of the single cell trajectories of the single 6, single 15, and four 1.5-minute pulses. Since Fig. 2 showed an increase in the median AUC of nuclear RelA (fold change) between the single 6-minute and 4×1.5-minute pulses and between the single 6-minute pulse and single 15-minute pulse, the emergent property score was defined as follows:

$$\text{Emergent Property Score} = abs\left(\frac{med(\mathbf{AUC_{4x1.5}})}{med(\mathbf{AUC_{1x6}})} - \frac{med(\widehat{AUC}_{4x1.5}(\boldsymbol{\theta}))}{med(\widehat{AUC}_{1x16}(\boldsymbol{\theta}))}\right) + abs\left(\frac{med(\mathbf{AUC_{1x15}})}{med(\mathbf{AUC_{1x6}})} - \frac{med(\widehat{AUC}_{1x15}(\boldsymbol{\theta}))}{med(\widehat{AUC}_{1x16}(\boldsymbol{\theta}))}\right) \tag{14}$$

Where $\widehat{AUC}_{1x6}(\boldsymbol{\theta})$, $\widehat{AUC}_{1x15}(\boldsymbol{\theta})$, $\widehat{AUC}_{4x1.5}(\boldsymbol{\theta})$ represents the data set of single-cell predictions for the parameter set $\theta$ for the AUC of the nuclear RelA (fold change) for the 1x6, 1×15, and 4x1.5-minute pulses. $AUC_{1×6}$, $AUC_{1×15}$, and $AUC_{4x1.5}$ represent the area under the curve for the experimental single cell trajectories.

**Model simulations**

D2FC and D2FC$^2$ were built using MATLAB 2023a. Associated code is available through the supplement (Supplementary Data 2) and the following repository (https://github.com/recleelab/D2FCSquared/, also on Zenodo https://doi.org/10.5281/zenodo.15977569). SimBiology

was used to generate ordinary differential equations from the reaction rates. Models were simulated for 10 days without cytokine stimulation to ensure a steady state was reached. Then the IKK profile corresponding to the appropriate dose was simulated to get the simulated results. Particle swarm optimization from MATLAB's Global optimization toolbox was used to optimize the parameter set to fit the average results from four scenarios used for fitting which included the control, 1×6 minute pulse, 1x30-minute pulse, and 4×1.5-minute pulse of IL-1. The global optimization attempted to minimize the following equation where $y_{i,t}$ is the experimental measurement for scenario i at time = $t$, and $\hat{y}_{i,t}(\theta)$ is the model estimate for parameter set $\boldsymbol{\theta}$:

$$J(\boldsymbol{\theta}) = \sum_i \sum_t \left(y_{i,t} - \hat{y}_{i,t}(\boldsymbol{\theta})\right)^2 + NFkB_{Herusitic}(\boldsymbol{\theta}) \tag{15}$$

Where:

$$NFkB_{Herusitic}(\boldsymbol{\theta}) = \begin{cases} 0 \text{ if } 0.01 & < R_{\frac{nuc}{cyt}} < 0.3 \\ 100 \text{ if } R_{\frac{nuc}{cyt}} & < 0.01 \\ 5 * R_{\frac{nuc}{cyt}} \text{ if } & R_{\frac{nuc}{cyt}} > 0.3 \end{cases} \tag{16}$$

The $NFkB_{Herusitic}$ was designed to ensure proposed parameters had a basal nuclear NF-κB described as the ratio of NF-κB in the nucleus to cytoplasm ($R_{\frac{nuc}{cyt}}$) within experimental expectations. Model values going below the lower limit were strongly penalized whereas models above were penalized proportional to ratio. The heuristic was set up to significantly penalize the overall optimization score if the basal nuclear NF-κB conditions were not met.

Particle swarm optimization was run with a swarm size of 100 particles and ceased after 20 consecutive iterations to prevent stalling, where the system could not achieve a reduction in the optimization score. Parameter values were optimized in log$_{10}$ format to help the optimizer explore the full parameter space across multiple orders of magnitude. The prior distribution of parameters ranges for optimization were carefully selected to represent biophysically relevant ranges around each parameter in the D2FC model (Supplementary table 4).

**Alterative model architecture**

Two alternative model architectures were independently tested, each extending the D2FC model with a distinct negative feedback mechanism involving either IκBβ or IκBε. The implementation of both feedback regulators followed prior modeling approaches[43,55]. Each inhibitor was modeled to bind NF-κB using the same structure as IκBα, but with its own set of kinetic parameters. IκBβ expression was modeled as constitutive and independent of NF-κB activation (Supplementary Table 5). In contrast, IκBε was treated as an NF-κB–inducible gene using the same transcriptional model as IκBα, but with a 45-minute transcriptional delay[54] (Supplementary Table 6). This delay was implemented using a Hill function that modulated the rate of transcription as a function of NF-κB concentration:

$$Rate_{IκBε, transcription} = V_{IκBε}(NFkB) \frac{t^{n_{delay}}}{t^{n_{delay}} + K_{delay}^{n_{delay}}} \tag{17}$$

Where $V_{IκBε}(NFkB)$ is the rate of IκBε transcription that is inducible by NF-κB expression and is multiplied by the time delay that is modeled based off of a hill equation dependent upon time where t is time in seconds, $n_{delay}$ is the hill coefficient, and $K_{delay}$ is the time delay of IκBε transcription start. Models were fit using particle swarm optimization exploring parameter ranges as defined in supplementary table 7 and 8.

**Quantification of zero- vs. first-order nuclear export kinetics**

Single-cell trajectories of nuclear NF-κB fold change were classified as having first- or zero-order nuclear export kinetics by fitting two models

to data at the point of nuclear NF-κB exiting the nucleus. A linear model was fit between the point of decay and the last data point. The first-order decay model was fitted by setting $N_0$ to the value of decay and finding the k value that minimized the sum of squared error. To avoid parameters for the first order decay rate that are linear, a minimum parameter exponential rate constant of k was set to 0.0167 min$^{-1}$, which represents a first order decay rate where approximately 95% of the nuclear Rela would exit the nucleus within 180 minutes. Defining a maximum value was essential. Setting this lower bound was critical for distinguishing between genuine first-order and prolonged pseudo-zero-order kinetics, simplifying our classification and improving interpretability of the export dynamics.

Zero-Order Decay Model:

$$y(t) = m * t + b \qquad (18)$$

First-Order Decay model:

$$y(t) = N_0 * e^{-k*t} \qquad (19)$$

For each single-cell trajectory, the time point of nuclear NF-κB exit rate was identified and the first and zero order models were fit to the data, only considering points after NF-κB started to exit the nucleus. The sum of squared error (SSE) was determined for both models and the nuclear RelA exit rate score is calculated to be the difference between the $SSE_{Zero} - SSE_{First}$. Positive values indicate zero-order and negative values indicate first-order nuclear export kinetics.

### Benchmarking the quality of the single cell predictions

Single-cell IKK trajectories were used as input for the model to generate predictions for each cell. The sum of squared errors (SSE) was calculated for each of the 194 single-cell model predictions to assess their accuracy. To classify the quality of these predictions, we established thresholds for 'excellent,' 'high,' and 'poor' fits. The threshold for 'high' fits was determined using an elbow plot. This was done by gradually increasing the threshold and counting the number of unacceptable fits (see Supplementary Fig. 8). To identify the elbow point, we first drew a line connecting the first and last points on the plot. We then measured the perpendicular distances from each point on the elbow plot to this line. The point with the maximum distance from the line was selected as the elbow point, which defined the optimal threshold for high-quality fits. To further classify the single cell trajectories, we divided the single cell trajectories between simulations that were considered excellent (which was defined as half of the elbow point) and high-quality fits.

### SimBiology model

A mechanistic model was constructed using SimBiology (MATLAB) by employing command-line tools within an m-file, rather than the graphical interface. This approach enhanced flexibility and automation in model development and simulation. SimBiology was used as a tool to automate the process of converting from reaction rate equations to a set of ordinary differential equations (ODE).

### Simulated CI Trajectories

Simulated CI trajectories were generated by first creating a zeros array with one value set to 1, representing the peak time. The array had a length of 46, matching the number of timepoints in our time-lapse imaging. The peak was positioned at 28 (index position 7) minutes, and a linear interpolation filled in the values between timepoints 0–28 and 28 to a variable adaptation time, resulting in trajectories with a triangular shape. To aid downstream fitting to Gaussian equations, a rolling average with a window of 3 was applied to smooth the trajectory. The smoothed CI trajectories were then fit to the sum of four Gaussian equations

for simulation in the model. To ensure the simulation started at zero, additional weight was added to t = 0 to the optimization function. With the method, we can precisely control the peak height, peak time, and adaptation time for the CI trajectories.

### Simulation of chromatin opening

To estimate the effect of chromatin permissiveness on the fraction of free NF-κB, a sub-model was constructed that included only the binding and unbinding of NF-κB to DNA. In this model, only free and bound NF-κB were dynamic, while all other species, specifically IκBα, were held constant at the steady state value before CI formation. The rate of NF-κB binding to DNA was determined by the equation:

$$\text{Rate}_{\text{Binding}} = \text{ka1d} * \text{DCoop} * \text{NPio}_{\text{const}} * [\text{N.NFkB}] \qquad (20)$$

where $\text{NPio}_{\text{const}}$ was held constant during the simulation but varied for generating the heatmap in Fig. 8c (bottom-right). The rate of NF-κB unbinding from DNA was influenced by both the basal dissociation rate and binding to IκBα. IκBα was set to a constant value, equal to the steady state value, $\text{N.IkBa}_{\text{steady-state}}$, after 10 days of simulation, to ensure the sub-model closely resembled the full model. Parameters from the best-fitting D2FC2 model were used for the simulations.

$$\begin{aligned}\text{Rate}_{\text{unbinding}} = \text{kd1d} * \left[\text{N.NFkBDNA}\right] \\ + \text{ka2a} * [\text{N.IkBa}_{\text{steady-state}}][\text{N.NFkBDNA}]\end{aligned} \qquad (21)$$

### Reporting summary

Further information on research design is available in the Nature Portfolio Reporting Summary linked to this article.

## Data availability

The single cell time courses and data used for plots generated in this study are provided in the Source Data and Supplementary Data files. Source data are provided with this paper.

## Code availability

The code used to develop the model, perform the analyses and generate results in this study is publicly available and has been deposited in the GitHub repository[82] https://github.com/recleelab/D2FCSquared and on Zenodo: https://doi.org/10.5281/zenodo.15977569 under the MIT license. The specific version of the code associated with this publication is archived in the Supplementary Data 2 file associated with this article.

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

## Acknowledgements

We thank other members of the Lee lab and colleagues in the Department of Computational and Systems Biology for many helpful discussions. This work was supported by generous funding to R.E.C.L. from NIH grant R35-GM119462.

## Author contributions

S.W.S., C.S.M. and R.E.C.L. designed the experiments; S.W.S., C.S.M., A.H.K. and P.M.C. performed the experiments; S.W.S., C.S.M. and R.E.C.L. analyzed data and developed models. S.W.S. and R.E.C.L. wrote the paper with edits from the remaining authors.

## Competing interests

The authors declare no competing interests.
