## [Transparent Peer Review file · Nature Communications]

Time-varying stimuli that prolong IKK activation promote nuclear remodeling and mechanistic switching of NF- κ B dynamics

Corresponding Author: Professor Robin Lee

Version 0:

Reviewer comments:

Reviewer #1

(Remarks to the Author)

Smeal et al. investigated how time-varying cytokine signals, specifically pulsatile IL-1 stimulation, are encoded by the NF- κ B signaling pathway in the U2OS cell line. This study highlights the role of NF- κ B-induced chromatin remodeling and subsequent DNA binding in the encoding process. Employing a dual-reporter cell line, the authors monitored NEMO spots and nuclear RelA to represent the input and output of the NF- κ B pathway, respectively. By applying pulsatile IL-1 stimulation through a previously developed microfluidic device (PMID: 31446223), they demonstrated that the adaptation time of NEMO spots, rather than the AUC, better correlates with the non-monotonic nuclear RelA AUC responses to pulsatile IL-1 stimulations. They further observed that nuclear RelA export shifts from first-order to zero-order kinetics, a process that also correlates nicely to the adaptation time of NEMO spots. Inspired by this observation, the authors refined a previously established mathematical model (PMID:24530305) to include a positive feedback loop of NF- κ B-induced chromatin opening and subsequent DNA sequestration of NF- κ B. This feedback depletes free NF- κ B, creating the conditions for zero-order kinetics. By optimizing the model parameters, they successfully captured the input-output relationship, the non-monotonic responses of nuclear RelA AUC, and the nuclear RelA export kinetic switch. Additionally, their model captures the RelA signal in response to IL-1 stimulation after chromatin relaxation with azacytidine treatment. Based on these results, the authors concluded that the NF- κ B pathway encodes information from pulsatile IL-1 stimulation through a mechanistic switch from first- to zero-order kinetic governed by prolong NEMO spots adaptation time and the positive feedback of NF- κ B sequestration by DNA.

This study aimed to resolve the underlying mechanisms through which pulsatile cytokine stimulations encode information into NF- κ B dynamics. While several studies have investigated NF- κ B dynamics and its underlying mechanisms in response to time-varying cytokine stimulation (PMID: 34211635 and 27381163), this study applied higher frequency pulsatile cytokine stimulations and revealed a novel phenotype featuring a mechanistic switching in nuclear RelA export. In addition, they integrated two mechanisms 1) transcription factor sequestration by its binding sites (PMID: 34568782 and 33305173), and 2) chromatin remodeling by NF- κ B (PMID: 24086160 and 34029641) to explain the observed shift in nuclear RelA export kinetics.

While the proposed mechanism could offer valuable insights into NF- κ B dynamics, the provided experimental evidence is preliminary and does not fully support their proposed mechanisms. Furthermore, the theoretical basis for how DNA sequestration explains the mechanistic switching is unclear. Here, we list several critical points for the authors to address before the study to be considered for publication.

1. One major mechanistic insight of this study is the DNA sequestration of NF- κ B for the conversion of nuclear RelA export kinetics from first-order to zero-order. To support that different IL-1 pulse dynamics (e.g., 1X6 min v.s. 4X1.5min) can lead to different NF- κ B binding mode (lower binding site accessibility v.s. higher binding site accessibility) and export kinetics (first-order v.s. zero-order), it is important to perform a NF- κ B ChIP-seq and a DNase-seq analyses after time-varied IL-1 stimulation.

2. Using azacytidine treatment, the authors wanted to demonstrate the DNA sequestration of NF- κ B can occur with the modification of chromatin compactness. However, their measurements are indirect, i.e., the basal level of nuclear NF- κ B and its kinetics, instead of its genomic occupancy. To directly monitor the genomic occupancy of NF- κ B after azacytidine

treatment, the authors need to carry out a NF- κ B Chip-seq analysis.

3. If there is a causality between chromatin compactness and nuclear RelA export kinetics using the azacytidine treatment, one would expect the conversion of the RelA export kinetics from first-order to zero-order with 1X6 min IL-1 treatment plus azacytidine as carried out in figure 6. We suggest the authors to quantify the percentage of zero- and first-order RelA-exporting cells with or without azacytidine treatment.

4. Azacytidine is known to be toxic to proliferating cells. For the experiment in figure 6D, azacytidine was treated up to 6 days. There is a concern about the off-target effect of the azacytidine that causes the drastic chromatin difference between the 3-day and 6-day treatments. Two ways to exclude this concern are 1. no cell death after 6-day azacytidine treatment and 2. gradual/graded changes in the kinetics of nuclear RelA 3 days, 4 days, 5 days and 6 days after azacytidine treatment.

5. Motivated by the observation on zero-order nuclear RelA export, the authors decided to investigate the DNA sequestration mechanism. They hypothesized that the zero-order kinetics of NF- κ B export is caused by the limited number of free NF- κ B molecules. However, the authors didn't provide adequate explanations or references to justify their hypothesis. According to the Hill equation, for a reactant to exhibit zero-order kinetics, the concentration of this reactant usually needs to be much higher than its binding partner, i.e., reaching saturation. Accordingly, NF- κ B concentration would be much higher than I κ B α , which is contradictory to the authors' hypothesis. We wonder if the author can provide a detailed reasoning for how limited free NF- κ B molecules can lead to zero-order kinetics of its export.

6. In the third section of the results (i.e., stimulation patterns that prolong NEMO puncta shifts the mechanism of NF- κ B nuclear export), the authors pointed out that a longer adaptation time of NEMO signal shifted the nuclear RelA export from first- to zero-order kinetics, enhancing the AUC of the nuclear RelA dynamics (figure 2A). However, we have concerns regarding to their definition of zero- and first-order kinetics. In figure 3D, the authors defined that zero-order kinetics of RelA export exhibit a lower decay rate. Consistently, the authors set a minimal decay rate in the first-order model in their classification (methodology section 8: Quantification of zero- v.s. first-order nuclear export kinetics). Accordingly, the slower decay kinetics (thus higher AUC) are more likely to be classified as zero-order in figure 3E, 3F and S4. We think that setting a minimal decay rate for the first-order model is a bias. Zero-order kinetics only suggest a constant decay rate, not necessarily faster or slower than the first-order kinetics. This potentially biased classification can lead to the positive correlation shown in figure 3F (between RelA exit rate score and adapt NEMO) considering the positive correlations observed in figure 2B (middle panel, between AUC and IL-1 pulse#) and figure 3C (between IL-1 pulse# and adapt NEMO). It would be more persuasive if the authors could obtain similar results without imposing a minimal decay rate in the first-order model to support mechanistic switching in export kinetics.

7. An alternative explanation for the difference in nuclear RelA kinetics between 1X6min v.s. 4X1.5min IL-1 stimulation observed in figure 1A can simply be a slower net nuclear export (but may still be a first-order decay) in 4X1.5min IL-1 stimulation, instead of a conversion in export kinetics from first-order to zero-order. To definitely know the decay kinetics after 4X1.5min IL-1 stimulation, it is necessary to extend the imaging time for RelA export for > 160 min.

Minor points:

1. In figure 5, the authors claimed that D2FC2 model recaptures the zero- and first-order kinetics. It will be better if the authors also show the quantification of the percentage of each kinetics in the population in simulations and compared the simulated results with that of the experimental results (figure 2 & 3).

2. Same as figure S5, same metric (the percentage of the zero- and first-order kinetics) can be applied to D2FC model and collectively show that D2FC2 model does recapture the mechanistic switching better.

3. In figure 6C, the type of statistic test applied is not mentioned in the legend.

4. For figure 7A, the three arrows pointing from the NEMO spots curve to the three stages of the chromatin remodeling diagram is confusing. Are they corresponding to the chromatin state and free NF- κ B at different time after IL-1 treatment?

5. For figure 7C, is there a reason that free nuclear NF- κ B fraction (the bottom right panel) is colored yellow instead of red? The readers may be confused with the first order label in the upper right panel.

6. For the methodology section of "Mechanistic modeling", the equation at line 621 should be IKKSpots(t) not IKK(t), and so is the IKK in the paragraph. Furthermore, for the Gaussian distribution function in the same equation, x should be substituted as t, if the authors were trying to fit the time course of NEMO spots with superposition of four Gaussian distribution functions.

7. For the methodology section of "Quantification of zero- v.s. first-order nuclear export kinetics", it will be better if the left-hand side of the equations of the two model can be unified as either y(t) or y.

8. What is the difference between zero order v.s. pseudo-zero order?

(Remarks on code availability)

Reviewer #2

(Remarks to the Author)

Growth factors of various sorts, including interleukins, are metabolically unstable and consequently, their release and degradation could result in cells having receptors for these hormones and responding to them, experience pulses, rather than boluses of stimulation. Smeal et al. asked how IL-1-sensitive cell would respond to such multiple pulse stimuli. As their model they chose U2OS cells with CRISPR/Cas9 edited NEMO to create a fusion with GFP to monitor input as formation of signaling “assemblies” as puncta in IL-1-stimulated cells and transcription factor NF- κ B subunit RelA fused to RFP for output readout as nuclear localization and kinetics of nuclear exit of RelA. Single cell image-based temporal quantification of NEMO-GFP puncta number and RelA nuclear intensity were measured. Initial studies of bolus treatments of cells with IL-1 for different periods of time showed, on average, linearly increasing responses with time. However, when the single bolus stimulation was replaced with a series of pulses of equal durations, this simple linear relationship between input and output broke down, and total signal for RelA nuclear localization increased above the average bolus response, with the number of pulses. They showed that this response was the result of a shift in nuclear export rate decay of RelA-RFP from first-, to zero-order. They attempted to account for these behaviors with a mathematical model (D2FC) but showed poor parameter fits based on a Particle swarm optimization strategy. Better fits resulted, however, if the effects of chromatin relaxation and thus increased binding sites for NF- κ B are accounted for, or both average and individual cells. Finally, the authors tested whether experimental relaxation of chromatin by treating cells with azacytidine, cytidine analog that incorporates into DNA and inhibits DNA-methyltransferase DNMT1, decreasing cytidine methylation. The resulting RelA nuclear localization signal diminished in a manner consistent with the modified D2FC-2 model.

The reported experiments and modelling studies appear to be well executed and the presentation of results, both written and figures, very clear illustrative. My one question is whether a more parsimonious model could result from taking into account recent results suggesting that the signaling assembly, and more specifically NEMO-polyubiquitin-mediated condensates by phase separation (Du, et al., Mol. Cell, 2022, doi: 10.1016/j.molcel.2022.03.037). The shift from first- to zero-order kinetics of RelA nuclear exit could be accounted for by maintenance of signaling condensates due to their incomplete first order disintegration by repetitive pulses of IL-1, resulting in keeping condensate component protein saturation concentrations above the phase boundary for formation of the condensates and thus a persistent signal. This is consistent with the authors observation that fewer pulse numbers and longer inter-pulse gap lengths tend to have weaker nuclear RelA responses (Supplementary Figs. 2 and 3).

To test the hypothesis that the shift in RelA export kinetics from first to zero order is caused by NEMO phase separation, I suggest that the author's perform the same mutations of NEMO, described in Du, et al., Fig. 5, in which they observed increases in the saturating concentration of NEMO. The predicted result would be that this would shorten the time between pulses that nuclear RelA response would weaken.

I think that these additional experiments would substantially add to an already interesting study by possibly revealing a novel and unexplored consequence of phase separation to signaling.

(Remarks on code availability)

Reviewer #3

(Remarks to the Author)

Smeal et al. 2024 Nature Comms review

Overall

The authors use an EGFP-NEMO, mCherry-RelA system that they previously published to quantify single cellular dynamics. Here they perform pulsatile inputs rather than sustained inputs, using a published microfluidic system. They find the cells respond more strongly than expected to repeating pulses of input. They perform model fitting to reveal the mechanisms of this effect with separated training and validation datasets.

Major

- Substantial literature in this area is omitted from both the introduction and the abstract.
 - o The statement on limited understanding of encoding and decoding on line 96 downplays substantial literature.
- Cheong et al., Science, 2011, Selimkhanov et al., 2014, Science, and particularly Adelaja et al., 2021, Immunity, must be cited and discussed here.
- Pulsatile inputs were investigated in Turner et al., J Cell Sci, 2010, which must be discussed here.
 - o Naigles et al., JBC, 2023 revealed the interplay of pulsatile inputs to inflammatory signalling and gene expression, including highlighting the importance of chromatin-opening dynamics.
- Do graphs of input pulses (such as 1D, 2A etc) represent the robot's inputs or a quantification of the dye+cytokine in the chamber? One might expect it takes some seconds (minutes?) for the cytokine to leave the chamber. The authors include a dye with the medium and cytokine to enable monitoring of cytokine in the chamber. This could be imaged/quantified fairly straightforwardly. Given the paper's focus on signal decay, knowing the decay of color in the cell culture channel would aid

interpretations of results. The authors should provide quantification of dye intensity in the cell culture channel in place of, or in addition to, input figures such as 1D and 2A.

The unintuitive results of Figure 2A could be explained entirely by the cytokine not being instantly removed and instead having a very small delay in “washing out” after each pulse. As the cells behave as if more pulses give you increased area under the curve despite the authors claim of AUC-preserving inputs. Plotting dye intensity in the cell chamber would resolve this potential alternative explanation. Imperfect switching off of the input would also explain why fewer pulses or longer gaps reduce the effect.

- The physiological relevance of extremely short pulse stimuli is unclear and somewhat overstated in the article. The manuscript states that in vivo TNF and IL-1 are cleared within minutes. The references indicate approximately 20 minutes for TNF and 3 minutes to distribute and 41 minutes to eliminate IL-1Beta. Other studies give peaks at 10 minutes but sustained presence of stimuli in circulation and tissues for hours. (Newton RC, et al. Lymphokine Res. 1988). The study of Oyler-Yaniv used to further justify the relevance of short pulses used 5h pulses. While the experimental set up is interesting to probe the extreme limits of the signalling axis, the physiological relevance is tenuous. Please re-write to remove this justification and instead focus on the study as testing the limits of NF-kB signalling.

- Figure 2C is presented to indicate a loss of correlation between NEMO and RelA (compared to 1F). However, the control condition (no stimuli) is only included in figure 1 and appears essential for the correlation. Control conditions should either be included or excluded in both Figure 1 and Figure 2C.

- The authors reveal t-adapt is key. It seems that for pulsatile stimuli the time that the final input ends is fairly predictive of the time that NEMO spots reach basal levels, and therefore AUC of RelA. The results could be summarised as: while NEMO is > 5 spots RelA decays with zero order kinetics, else it declines with first-order kinetics. Put simply perhaps a more simple explanation than the one explored by the paper is that if there are any C1 assemblies, there is sufficient NEMO for slow NF-kB decay. This explanation doesn't require epigenetics, therefore weakening the motivation for the rest of the paper.

- The D2Fc model used for particle swarm optimization does not contain many negative feedback loops that may impact decay of nuclear NF-kB = I κ Bbeta, epsilon, p100/I κ B delta. This is particularly important given the conclusion of missing topology. While adding additional topology, focused around DNA binding improved model fits, no other model architecture was investigated. Could the authors investigate whether adding additional feedback loops with different timescales/kinetics could explain the data? For example p100/I κ Bd and I κ Be.

- The only experimental validation of chromatin accessibility being the correct mechanism to explain the emergent behavior of cells is an experiment in which azacitidine causes increased RelA activity and decreased RelA induction. Due to the quantification of RelA induction as fold changes, it is somewhat inevitable that increased nuclear activity at basal will result in reduced fold changes. Therefore, the main prediction is increased RelA at basal.

In a different cell line Azacitidine has been previously shown to inhibit nuclear RelA at 1, 2 and 3 days, the authors find no change at 3 days and an increase at 6 days (Khong et al., 2008, Haematologica) suggesting the results presented here are cell-type specific rather than universally applicable.

An alternative hypothesis is that RelA is induced through another mechanism. For example, cell death in the cell culture which occurs in U2OS cells treated with DNA methylation inhibitors (Al-Romaih et al., cancer Cell International 2007).

I would have thought that if nuclear retention on DNA is key to the observed dynamics then this model would decouple NEMO and RelA dynamics, such that increased nuclear RelA is present without increased NEMO. Is that seen in this model? Is that seen in the data in response to azacytidine. Without this test there are too many alternative hypotheses that explain the data. Show this data.

Minor

- Line 57-60 suggest that stochasticity is the only source of variability. There is substantial variability can be explained deterministically with sufficient characterisation (Roy et al., 2018, PNAS).

- Line 88 suggests only RelA fusion is available but recently reported cRel fusion proteins should be mentioned (Rahman et al., 2024, Cell Reports)

- Consider adding a fitting line to Fig 1F

- Ensure the Greek kappa is used throughout

- Why U2OS cells here? A lot of previous data in MEFs is not comparable to the data generated here as a result of the different cell system. Similarly many existing models have been fitted to MEF data but are not used here. Why?

(Remarks on code availability)

Reviewer #4

(Remarks to the Author)

(Remarks on code availability)

Version 1:

Reviewer comments:

Reviewer #1

(Remarks to the Author)

During the first review, we emphasized the need to directly demonstrate that DNA-mediated sequestration of NF- κ B underlies the switch in its nuclear-export kinetics. In the authors responses, they combined FRAP and HeK4me3 immunofluorescence to address our major concern. Figure 6 shows that pulsatile IL-1 stimulation increases the immobilized NF- κ B through FRAP, indicating increased NF- κ B binding. Figure 7 reveals a parallel rise in H3K4me3, a modification that opens chromatin, under the same stimulation. These complementary data align with their proposed mechanisms, and strengthens the argument that the interaction with DNA modulate the NF- κ B dynamics.

We appreciate the authors' efforts to address our concerns. However, we think a key mechanistic link between the results of figure 6 and 7 is still missing. The observed increase in immobilized NF- κ B and H3K4me3 under pulsatile IL-1 stimulation are correlative. To establish the causality, we recommend the authors to directly reduce H3K4me3 by inhibiting relevant methyltransferases, for example SET1 or MLL1 through knockdown or small molecules such as MI-501 (PMID: 25817203) or OICR-9429 (PMID: 34154613). Through these interventions, the authors could examine whether the immobilization of NF- κ B will be reduced or the export kinetics under pulsatile IL-1 stimulation will be reversed.

In addition to the suggestions above for deepening mechanistic insight we have several suggestions to improve the data presentation.

Figure 6

1. In panel B, the authors showed an example image. It will be more convincing to show example images for both 1X6min and 4X1.5min stimulations.
2. In panel C, the legend describes the traces as example trajectories, but the curves are unusually smooth. Please clarify whether any filtering or processing was applied?
3. The total immobilized NF- κ B level in panel C (around 3 units) and panel D (up to 100 units) are quite different. Please explain the source of this discrepancy?

Minor point

1. The "Simulation of Chromatin Opening" section in the methodology still mentions azacitidine experiment, which has been removed from the manuscript. Please revise this section according to the new manuscript.

(Remarks on code availability)

Reviewer #2

(Remarks to the Author)

The authors have adequately addressed my questions and I now recommend publication without further changes

(Remarks on code availability)

Reviewer #3

(Remarks to the Author)

The authors have made strong efforts to answer my comments from the previous review.

The graphs from high frequency imaging experiments, and AUC-controlled data are compelling and greatly strengthen the study. The removal of the AZA data has also better focused the study on the most compelling data.

The authors admit that the drug-response data was not the strongest, and that the physiological relevance was understated. Therefore the impact of this study beyond this experimental system remains somewhat unclear and the overall readership of the manuscript now that it is focused on the key data may be somewhat smaller.

Despite this, I believe the manuscript makes a substantial contribution to the field and will be of interest to many reader in Nature communications.

(Remarks on code availability)

Reviewer #4

(Remarks to the Author)

(Remarks on code availability)

Version 2:

Reviewer comments:

Reviewer #1

(Remarks to the Author)

The authors have addressed my questions and I now recommend publication.

(Remarks on code availability)

Dear Reviewers:

We would like to thank you for taking the time to provide insightful and highly constructive comments. Our efforts to address all the concerns raised by the first submission are described below in the point-by-point response. We hope the reviewers will agree that by addressing their comments our revised manuscript has improved significantly, and we hope that it will be considered acceptable for publication in *Nature Communications*. In the revised manuscript, textual changes are marked in blue font. Here we provide a brief overview of alterations to figures and new results provided in additional figures:

We added significant experimental evidence to support the core assumptions of the D2FC² model which included: FRAP to demonstrate the predicted increase in NF- κ B immobilization on DNA, and immunofluorescence staining of H3K4me₃, a marker of open chromatin and active transcription (Figure 6 and 7). To better highlight our more direct results we opted to remove the model predictions and experiments for Azacytidine responses and NF- κ B dynamics but would be open to re-integrating data should the reviewers request this change. We also tested two alternative models that added I κ B ϵ or I κ B β negative feedback mediators found that the D2FC² model better recapitulated the emergent property reported in this manuscript (see supplementary figure 7).

Removed Figures:

- Figure 6 and supplementary figure 8, which contained the azacytidine analysis was removed in favor of more direct experimental evidence which directly measures the immobilized fraction of nuclear Rel-A (new figure 6) and the H3K4me₃ modifications (new figure 7) to a cytokine pulse train.

Summary of figure panel changes:

For clarity, all figure numbers refer to their updated designations in the revised manuscript. References to the old figure labels will be included when appropriate

- In Figure 1, panel F, the control (no stimulation) data were excluded from both the plot and the Spearman rank correlation calculation to avoid artificially grounding the correlation.
- In figure 5d, we originally showed the quality of the single cell model predictions of the nuclear RelA between each pulse pattern and how the aggregate single cell predictions across the top 10 D2FC² parameter sets. We opted to update the figure by including example trajectories of the different model classification systems (previously in supplementary figure 7A) and moving the single cell model predictions of the overall best model to the supplementary text to better illustrate that our single cell predictions do not depend on one single parameter set.
- New figure 6 includes model predictions for amounts of NF- κ B bound to DNA with experimental validation using fluorescence recovery after photobleaching (FRAP). Results confirm model prediction that DNA-binding is enhanced in 4x1.5-minute at early and late time points.
- New figure 7 includes model predictions for chromatin accessibility with experimental validation using fixed cell immunofluorescence against H3K4me₃. Results confirm model prediction that chromatin accessibility is only enhanced in 4x1.5-minute at later times.

- Figure 8 included with the following changes to the panels:
 - o Panel A: the schematic was updated to better convey that the dynamics of the NEMO encoding features could change the chromatin state.
 - o Panel C (bottom right) color scheme was changed to be consistent with the other heatmaps to maintain consistency.
- Supplementary figure 1 includes two new panels (C and D) to highlight the consistency in the area under the curve for the single and multi-pulse train stimulation patterns.
- Supplementary figure 4 includes two new panels (D and E) to highlight the RelA export kinetics for 4x1.5min continue to follow zero order kinetics well after the NEMO punctate formation are gone.
- Supplementary figure 5 contains four new panels (D-G) to show how D2FC² model succeeds over the optimized D2FC to convert nuclear RelA exit rate from zero to first order dynamics between the 1X6 and 4X1.5-minute pulse trains.
- New Supplementary figure 7 highlights how the addition of either I κ B ϵ or I κ B β feedback mechanisms to the D2FC model does not recapitulate the emergent property of observed in our multi-pulse stimulation trains.
- New supplementary figure 9 contains supporting data for experimental validation of an increased immobilized fraction of nuclear RelA and increase in chromatin remodeling.

Reviewer #1 (Remarks to the Author):

Smeal et al. investigated how time-varying cytokine signals, specifically pulsatile IL-1 stimulation, are encoded by the NF- κ B signaling pathway in the U2OS cell line. This study highlights the role of NF- κ B-induced chromatin remodeling and subsequent DNA binding in the encoding process. Employing a dual-reporter cell line, the authors monitored NEMO spots and nuclear RelA to represent the input and output of the NF- κ B pathway, respectively. By applying pulsatile IL-1 stimulation through a previously developed microfluidic device (PMID: 31446223), they demonstrated that the adaptation time of NEMO spots, rather than the AUC, better correlates with the non-monotonic nuclear RelA AUC responses to pulsatile IL-1 stimulations. They further observed that nuclear RelA export shifts from first-order to zero-order kinetics, a process that also correlates nicely to the adaptation time of NEMO spots. Inspired by this observation, the authors refined a previously established mathematical model (PMID:24530305) to include a positive feedback loop of NF- κ B-induced chromatin opening and subsequent DNA sequestration of NF- κ B. This feedback depletes free NF- κ B, creating the conditions for zero-order kinetics. By optimizing the model parameters, they successfully captured the input-output relationship, the non-monotonic responses of nuclear RelA AUC, and the nuclear RelA export kinetic switch. Additionally, their model captures the RelA signal in response to IL-1 stimulation after chromatin relaxation with azacytidine treatment. Based on these results, the authors concluded that the NF- κ B pathway encodes information from pulsatile IL-1 stimulation through a mechanistic switch from first- to zero-order kinetic governed by prolong NEMO spots adaptation time and the positive feedback of NF- κ B sequestration by DNA.

This study aimed to resolve the underlying mechanisms through which pulsatile cytokine stimulations encode information into NF- κ B dynamics. While several studies have investigated NF- κ B dynamics and its underlying mechanisms in response to time-varying cytokine stimulation (PMID: 34211635 and 27381163), this study applied higher frequency pulsatile cytokine stimulations and revealed a novel phenotype featuring a mechanistic switching in nuclear RelA export. In addition, they integrated two mechanisms 1) transcription factor sequestration by its binding sites (PMID: 34568782 and 33305173), and 2) chromatin remodeling by NF- κ B (PMID: 24086160 and 34029641) to explain the observed shift in nuclear RelA export kinetics.

While the proposed mechanism could offer valuable insights into NF- κ B dynamics, the provided experimental evidence is preliminary and does not fully support their proposed mechanisms. Furthermore, the theoretical basis for how DNA sequestration explains the mechanistic switching is unclear. Here, we list several critical points for the authors to address before the study to be considered for publication.

We thank the reviewer for the thorough evaluation, their constructive ideas, and encouraging comments. Although we did not follow exactly the recommended course in preparing our revision, the suggestions helped shape our work, and believe the revised manuscript is improved dramatically with substantive validation of our proposed model.

1. One major mechanistic insight of this study is the DNA sequestration of NF- κ B for the conversion of nuclear RelA export kinetics from first-order to zero-order. To support that different IL-1 pulse dynamics (e.g., 1X6 min v.s. 4X1.5min) can lead to different NF- κ B binding mode (lower binding site accessibility v.s. higher binding site accessibility) and export kinetics (first-

order v.s. zero-order), it is important to perform a NF- κ B Chip-seq and a DNase-seq analyses after time-varied IL-1 stimulation.

We thank the reviewer for this suggestion, and we also considered sequencing-based experiments as validation approaches. Unfortunately, we decided against these for two reasons. First, the extremely high cost to perform these experiments correctly is prohibitive without dedicated funding and would likely lead to confirmatory results of the elegant study published recently in Science by others (PMID: 34140389). Second, sequencing assays require cellular material far greater than can be collected from our microfluidic cell cultures and would therefore require scaling the system in uncertain ways.

Instead, we opted to validate the core assumptions and predictions of the model in two ways, using FRAP to show predicted enhancement of NF- κ B immobilized on DNA and immunofluorescence of H3K4me3, a marker for open chromatin and active gene transcription. In validating in this way, we could directly compare outputs of our model with observables from experiments, and using the same instrumentation demonstrate the model's accuracy. We highlight to the reviewer that the model parameterization used for predictions have not changed since our prior submission, demonstrating that our model has provided mechanistic insights for how dynamic stimuli in regulation of CI can lead to distinct signaling responses and cellular states.

2. Using azacytidine treatment, the authors wanted to demonstrate the DNA sequestration of NF- κ B can occur with the modification of chromatin compactness. However, their measurements are indirect, i.e., the basal level of nuclear NF- κ B and its kinetics, instead of its genomic occupancy. To directly monitor the genomic occupancy of NF- κ B after azacytidine treatment, the authors need to carry out a NF- κ B Chip-seq analysis.

We agree with the reviewer that azacytidine treatment provides only an indirect assessment of relationships between NF- κ B, DNA binding, and chromatin dynamics. This sentiment was further emphasized by comments from Reviewer #3. Although we still feel that the concordance between model predictions and experiments for Azacytidine responses and NF- κ B dynamics are quite interesting, we decided to instead remove these data completely and focus on the more direct model validation experiments used here in the revised manuscript. We would be open to re-integrating these results should the Reviewer(s) suggest it meaningfully supports our findings in a way that is not distracting. Otherwise, we feel that the current validation results are more focused and that the azacytidine data could have a more detracting effect on the reader.

3. If there is a causality between chromatin compactness and nuclear RelA export kinetics using the azacytidine treatment, one would expect the conversion of the RelA export kinetics from first-order to zero-order with 1X6 min IL-1 treatment plus azacytidine as carried out in figure 6. We suggest the authors to quantify the percentage of zero- and first-order RelA-exporting cells with or without azacytidine treatment.

Our original hypothesis aligned with the reviewer's expectation—that RelA export kinetics would shift from first-order to zero-order under 1X6-minute IL-1 treatment following azacytidine exposure. However, our simulations (Panel A,B below, previous Figure 6 and now removed) predicted a different outcome: as chromatin accessibility increases, basal nuclear RelA levels rise

while the peak fold-change response to a single 6-minute pulse diminishes. These predictions were supported by experimental data. The unexpected finding was the pronounced increase in basal NF- κ B levels in response to azacytidine, highlighting a chromatin-mediated effect on nuclear NF- κ B dynamics that diverged from our initial expectations.

4. Azacytidine is known to be toxic to proliferating cells. For the experiment in figure 6D, azacytidine was treated up to 6 days. There is a concern about the off-target effect of the azacytidine that causes the drastic chromatin difference between the 3-day and 6-day treatments. Two ways to exclude this concern are 1. no cell death after 6-day azacytidine treatment and 2. gradual/graded changes in the kinetics of nuclear RelA 3 days, 4 days, 5 days and 6 days after azacytidine treatment.

During the experiment, control and treated cells were initially seeded at the same density on day 1 and cultured under identical conditions. On day 2, cells were counted and then re-seeded at equal numbers into a 96-well plate. At this point, the total cell numbers in both the azacytidine-treated and control groups were comparable, with the ratio of control to treated cells. By day 5, however, the ratio of control:azacytidine cell numbers increased sharply, ranging from 4 to 9-fold. This significant difference strongly suggests that azacytidine exerted a substantial effect on the cells. We suspect that reduced growth rate and cell death played a major role in the reduced number of cells following azacytidine. Given that azacytidine is likely inducing some amount of cell death,

we did observe (anecdotally) graded increases in the amount of the basal nuclear NF- κ B in the intermediate days between 3 and 6. We provide this discussion to the reviewer only as a point of interest, as the associated data have been removed from the revised manuscript.

5. Motivated by the observation on zero-order nuclear RelA export, the authors decided to investigate the DNA sequestration mechanism. They hypothesized that the zero-order kinetics of NF- κ B export is caused by the limited number of free NF- κ B molecules. However, the authors didn't provide adequate explanations or references to justify their hypothesis. According to the Hill equation, for a reactant to exhibit zero-order kinetics, the concentration of this reactant usually needs to be much higher than its binding partner, i.e., reaching saturation. Accordingly, NF- κ B concentration would be much higher than I κ B α , which is contradictory to the authors' hypothesis. We wonder if the author can provide a detailed reasoning for how limited free NF- κ B molecules can lead to zero-order kinetics of its export.

We thank the reviewer for this comment, as it highlighted a lacking clarity and prompted us to refine description of zero-order kinetics. We have modified the text throughout to add clarity to the combined mechanisms where the net effect is due to competition in the nucleus for NF- κ B binding to either DNA or proteins associated with negative feedback – importantly, as NF- κ B has more binding sites available, competition for generic and promoter specific protein-DNA interactions versus protein-protein interactions with negative feedback mediators increases.. These are in combination of all reactions with first order kinetics that appear zero-order (i.e. pseudo-zero-order) through their push-and-pull interplay. We hope the reviewer will agree that through edits and the additional validation results, this interplay has become clearer.

6. In the third section of the results (i.e., stimulation patterns that prolong NEMO puncta shifts the mechanism of NF- κ B nuclear export), the authors pointed out that a longer adaptation time of NEMO signal shifted the nuclear RelA export from first- to zero-order kinetics, enhancing the AUC of the nuclear RelA dynamics (figure 2A). However, we have concerns regarding to their definition of zero- and first-order kinetics. In figure 3D, the authors defined that zero-order kinetics of RelA export exhibit a lower decay rate. Consistently, the authors set a minimal decay rate in the first-order model in their classification (methodology section 8: Quantification of zero- v.s. first-order nuclear export kinetics). Accordingly, the slower decay kinetics (thus higher AUC) are more likely to be classified as zero-order in figure 3E, 3F and S4. We think that setting a minimal decay rate for the first-order model is a bias. Zero-order kinetics only suggest a constant decay rate, not necessarily faster or slower than the first-order kinetics. This potentially biased classification can lead to the positive correlation shown in figure 3F (between RelA exit rate score and adapt NEMO) considering the positive correlations observed in figure 2B (middle panel, between AUC and IL-1 pulse#) and figure 3C (between IL-1 pulse# and adapt NEMO). It would be more persuasive if the authors could obtain similar results without imposing a minimal decay rate in the first-order model to support mechanistic switching in export kinetics.

We understand the reviewer's concern and appreciate their suggestion to assess potential bias by recalculating the nuclear RelA exit scores without enforcing a lower bound on the first-order decay rate. Here, we demonstrate to the reviewer that for trajectories previously classified as first-order, removing the constraint of a minimal rate, had no effect on the calculated exit scores (see reviewer Figure 2 below). In contrast, for trajectories classified as zero-order, the nuclear RelA exit rate was

Reviewer Fig. 2: By removing the lower limit constraint on the zero-order decay rate, the first-order model can closely approximate zero-order behavior in the timescale of experiments, which in turn causes the nuclear RelA exit scores to approach zero.

constrained to values near zero because nuclear RelA exit dynamics consistent with zero-order behavior produced first-order model fits that were indistinguishable from those of the zero-order model (see Reviewer Figure 3 below). This is because first-order decay with very slow rates appear linear in the timescale of the experiments. Therefore, without having a limit on adaptation time for the first-order process via a minimal decay rate, zero-order processes becomes a special case of first order processes.

Reviewer Fig. 3: Example trajectories highlighting the exit rate score when removing the minimum rate. Top row show example fits of the first and zero order models on nuclear Rel-A trajectories displaying first order. Bottom row is showing data displaying zero order exit rate where both the zero-order and first-order model are indistinguishable.

This, effectively, compresses the dynamic range of the zero-order exit rates, resulting in a visually compacted region for the zero-order portion of the graph. Regenerating the graphs in fig. 3E,F does not yield significantly different p-values and does not change our conclusions since the monotonic relationship between the adaptation time and time to max to the nuclear rate exit scores remain unchanged (see Reviewer Figure 4 below). Fundamentally, our main conclusions are not affected by the presence or absence of the minimum rate, and we do not believe this is a bias but rather simplifies the classification and improves the interpretability of our data and results. dynamics. We updated lines 815-818 to clarify our position on setting a minimum rate. Secondly,

we more clearly state in the text our two-fold definition for pseudo-zero-order kinetics to explicitly describe the minimal adaptation time that is used to establish the minimal decay rate enabling

comparison of single-pulse and multi-pulse response dynamics.

7. An alternative explanation for the difference in nuclear RelA kinetics between 1X6min v.s. 4X1.5min IL-1 stimulation observed in figure 1A can simply be a slower net nuclear export (but may still be a first-order decay) in 4X1.5min IL-1 stimulation, instead of a conversion in export kinetics from first-order to zero-order. To definitely know the decay kinetics after 4X1.5min IL-1 stimulation, it is necessary to extend the imaging time for RelA export for > 160 min.

As the reviewer suggested, in new Supplementary Fig. S4d and e, we show data with an imaging time that has been extended to 5 hours and quantified the first order vs. zero order fits. These observations confirm that the nuclear RelA export kinetics for 4x1.5min continue to follow the pseudo-zero order pattern and classify as such. Relevant text in the manuscript has also been added on lines 286-290.

Minor points:

1. In figure 5, the authors claimed that the D2FC2 model recaptures the zero- and first-order kinetics. It would be better if the authors also showed the quantification of the percentage of each kinetics in the population in simulations and compared the simulated results with that of the experimental results (Figures 2 & 3).

We thank the reviewer for your great suggestion. See comment below for minor point 2.

2. Same as figure S5, same metric (the percentage of the zero- and first-order kinetics) can be applied to D2FC model and collectively show that D2FC2 model does recapture the mechanistic switching better.

We updated Supplementary Figure 5 to include bar charts showing the fraction of simulated trajectories with export kinetics that followed zero-order dynamics. As shown in Supplementary Figure 5d, and e the original D2FC model fails to replicate the experimental data. While the optimized D2FC model captures zero-order kinetics, it lacks the ability to switch between first- and zero-order export in response to different stimulation patterns, displaying zero-order kinetics across both single and multi-pulse inputs. In contrast, the D2FC2 model successfully recapitulates the experimentally observed switch from first-order export under a single 6-minute pulse to zero-order export under multi-pulse stimulation. Additionally, on line 395 we now directly reference the new figure in support of our following statement: “*Visually, simulated single-cell time courses of nuclear NF- κ B appeared like experiments, showing both first-order and zero-order nuclear export kinetics appropriate to each condition (Fig. 5a and Supplementary Fig. 5c).*”

3. In figure 6C, the type of statistic test applied is not mentioned in the legend.

Figure 6c in the original manuscript was part of the azacitidine analysis that has been removed in this revision. However, in the first submission of figure 6C, it was a student’s t-test.

4. For figure 7A, the three arrows pointing from the NEMO spots curve to the three stages of the chromatin remodeling diagram is confusing. Are they corresponding to the chromatin state and free NF- κ B at different time after IL-1 treatment?

The original intent of this figure 7A (now Figure 8A) was to convey that different dynamics of the EGFP-NEMO spot formation can lead to different binding site accessibility. In other words, the NEMO encoding features such as C_{max} and t_{adapt} could provide information about the chromatin state. We believe this edit has enhanced clarity.

5. For figure 7C, is there a reason that free nuclear NF- κ B fraction (the bottom right panel) is colored yellow instead of red? The readers may be confused with the first order label in the upper right panel.

We made the color a different scheme because it contained different axis and we wanted it to stand out. We understand this may introduce more confusion than it alleviates and we updated the color scheme to the bottom right panel for consistency.

6. For the methodology section of “Mechanistic modeling”, the equation at line 621 should be $IKKSpots(t)$ not $IKK(t)$, and so is the IKK in the paragraph. Furthermore, for the Gaussian distribution function in the same equation, x should be substituted as t , if the authors were trying to fit the time course of NEMO spots with superposition of four Gaussian distribution functions.

Thank you for careful reading of our methods section and the text has been updated accordingly.

7. For the methodology section of “Quantification of zero- v.s. first-order nuclear export kinetics”, it will be better if the left-hand side of the equations of the two model can be unified as either $y(t)$ or y .

Text has been updated to be consistent between the two equations.

8. What is the difference between zero order v.s. pseudo-zero order?

True zero-order reactions, where the rate is entirely independent of reactant concentration, are rare in nature. However, biological systems can exhibit apparent zero-order kinetics under specific conditions. A classic example is enzyme-substrate kinetics: when the substrate concentration significantly exceeds the enzyme's capacity (saturation), and the enzyme concentration is constant, the rate of substrate metabolism becomes constant and appears to follow pseudo-zero-order kinetics. This is because the enzyme is working at its maximum velocity. As the substrate is consumed and its concentration drops below the saturation point, the reaction kinetics will revert to the typical enzyme-substrate relationship (often Michaelis-Menten kinetics, which can approximate first-order at low substrate concentrations). Therefore, zero order rates in biology are not due to a reaction being truly independent of the concentration but a result of first order processes that buffer or saturate a process. We have updated lines 286-290 with the definition to clarify this point.

Reviewer #2 (Remarks to the Author):

Growth factors of various sorts, including interleukins, are metabolically unstable and consequently, their release and degradation could result in cells having receptors for these hormones and responding to them, experience pulses, rather than boluses of stimulation. Smeal et al. asked how IL-1-sensitive cell would respond to such multiple pulse stimuli. As their model they chose U2OS cells with CRISPR/Cas9 edited NEMO to create a fusion with GFP to monitor input as formation of signaling “assemblies” as puncta in IL-1-stimulated cells and transcription factor NF- κ B subunit RelA fused to RFP for output readout as nuclear localization and kinetics of nuclear exit of RelA. Single cell image-based temporal quantification of NEMO-GFP puncta number and RelA nuclear intensity were measured. Initial studies of bolus treatments of cells with IL-1 for different periods of time showed, on average, linearly increasing responses with time. However, when the single bolus stimulation was replaced with a series of pulses of equal durations, this simple linear relationship between input and output broke down, and total signal for RelA nuclear localization increased above the average bolus response, with the number of pulses. They showed that this response was the result of a shift in nuclear export rate decay of RelA-RFP from first-, to zero-order. They attempted to account for these behaviors with a mathematical model (D2FC) but showed poor parameter fits based on a Particle swarm optimization strategy. Better fits resulted, however, if the effects of chromatin relaxation and thus increased binding sites for NF- κ B are accounted for, or both average and individual cells. Finally, the authors tested whether experimental relaxation of chromatin by treating cells with azacytidine, cytidine analog that incorporates into DNA and inhibits DNA-methyltransferase DNMT1, decreasing cytidine methylation. The resulting RelA nuclear localization signal diminished in a manner consistent with the modified D2FC-2 model.

The reported experiments and modelling studies appear to be well executed and the presentation of results, both written and figures, very clear illustrative. My one question is whether a more parsimonious model could result from taking into account recent results suggesting that the signaling assembly, and more specifically NEMO-polyubiquitin -mediated condensates by phase separation (Du, et al., Mol. Cell, 2022, doi: 10.1016/j.molcel.2022.03.037). The shift from first- to zero-order kinetics of RelA nuclear exit could be accounted for by maintenance of signaling condensates due to their incomplete first order disintegration by repetitive pulses of IL-1, resulting in keeping condensate component protein saturation concentrations above the phase boundary for formation of the condensates and thus a persistent signal. This is consistent with the authors observation that fewer pulse numbers and longer inter-pulse gap lengths tend to have weaker nuclear RelA responses (Supplementary Figs. 2 and 3).

To test the hypothesis that the shift in RelA export kinetics from first to zero order is caused by NEMO phase separation, I suggest that the author’s perform the same mutations of NEMO, described in Du, et al., Fig. 5, in which they observed increases in the saturating concentration of NEMO. The predicted result would be that this would shorten the time between pulses that nuclear RelA response would weaken.

I think that these additional experiments would substantially add to an already interesting study by possibly revealing a novel and unexplored consequence of phase separation to signaling.

We thank the reviewer for the thoughtful feedback and kind words involving the execution and presentation of our results. Regardless of the mechanism underlying IKK spot formation, it is important to point out that our approach is agnostic to the mechanism of spot formation, whether condensates or other types of multi-protein assemblies.

The reviewer has asked if individual IKK spots are maintained in multi-pulse stimulation patterns, persisting for longer by slowing their decay. Importantly, although we do not track single spots here, we have done so in several other projects (including PMID: 34301608; biorxiv: <https://doi.org/10.1101/2025.04.18.649561>). From these single particle tracking experiments, we have seen consistently that the dynamics of NEMO recruitment at CI is highly stereotyped with per-complex adaptation times that are consistent in different concentration, dynamic environments, and even in response to IL-1-related ligands with different affinities (see biorxiv). That is to say, that dynamic stimuli are not maintaining CI assemblies/condensates for longer but instead distribute the timing of their formation over longer periods. Edits to the text attempt to emphasize these points.

We also appreciate the reviewer's suggestion of a compelling experiment to examine the role of NEMO's ubiquitin-binding domains in our observations. As highlighted by Du et al. (Mol. Cell, 2022; doi: 10.1016/j.molcel.2022.03.037), deletion of the NUB, ZF, or both domains significantly impairs NEMO's ability to undergo phase separation with polyubiquitin (Figure 5). Based on our published findings, we surmise that disrupting these domains would either dampen or completely

Reviewer Figure 5: 1,6 Hexanediol following IL-1 stimulation does not impact CI spots.

prevent the overall trajectory of IKK spot formation and decay, ultimately precluding nuclear RelA responses.

We appreciate the reviewer's important question regarding mechanisms of phase separation and their impact on signaling. In Reviewer Figure 5, we tested whether the observed IKK spots are phase-separated condensates within living cells by the criteria of liquidity disruption via 1,6-hexanediol (1,6-HD) that has been observed both *in vitro* and *in vivo*. We first stimulated cells with IL-1 followed by 0.89% or 1.77% 1,6-HD. Typically, these concentrations eliminate phase-separated condensates within two minutes of treatment (PMID: 34404453). However, in our cells the IKK spots remained visible after 5 minutes of 1,6-HD exposure even in cells undergoing 1,6-HD-induced death, suggesting the IKK spots may not be typical phase-separated droplets. Although this result does not unequivocally prove that the NEMO spots are not phase-separated under endogenous expression within living cells, it does suggest that there is work to be done in understanding their composition and dynamics of assembly and disassembly. We hope the reviewer agrees with us that this goes beyond the scope of the project presented here, and it is a topic that we will be pursuing separately. We bring up some of these points and perspectives in the discussion and acknowledge the related works in the literature on lines 592-600.

Reviewer #3 (Remarks to the Author):

Smeal et al. 2024 Nature Comms review

Overall

The authors use an EGFp-NEMO, mCherry-RelA system that they previously published to quantify single cellular dynamics. Here they perform pulsatile inputs rather than sustained inputs, using a published microfluidic system. They find the cells respond more strongly than expected to repeating pulses of input. They perform model fitting to reveal the mechanisms of this effect with separated training and validation datasets.

Major

1. Substantial literature in this area is omitted from both the introduction and the abstract.
 - The statement on limited understanding of encoding and decoding on line 96 downplays substantial literature. Cheong et al., Science, 2011, Selimkhanov et al., 2014, Science, and particularly Adelaja et al., 2021, Immunity, must be cited and discussed here.

We appreciate the reviewer's comment and, upon careful reflection, agree that our original manuscript unintentionally overlooked substantial literature on NF- κ B signaling, particularly in terms of information theory-related studies. Our intent was to highlight a more specific gap in understanding, how IKK recruitment to polyubiquitin chains transduces dynamic extracellular signals to downstream NF- κ B activation. We have revised line 96 to clarify this point and have incorporated the relevant references suggested by the reviewer to appropriately acknowledge these prior works (including our own). Please see lines 96-103. Subsequently in the discussion we relate the outlook back to information-based perspectives. We hope the reviewer will agree that this perspective enhances the manuscript.

- Pulsatile inputs were investigated in Turner et al., J Cell Sci, 2010, which must be discussed here.

The primary focus of this study was to investigate how SK-N-AS cells respond to low doses of TNF α (e.g., 10 pg/mL), showing that lower concentrations activate a smaller fraction of cells, as measured by nuclear RelA localization. They did report that fewer number of cells responded to a short 5-minute pulse compared to a continuous stimulation of TNF stimulation indicating that NF- κ B does not solely depend on ligand concentration but duration. We've included this in our motivation of single pulse experiments on lines 171-172.

- Naigles et al., JBC, 2023 revealed the interplay of pulsatile inputs to inflammatory signalling and gene expression, including highlighting the importance of chromatin-opening dynamics.

This was an interesting study where they Naigles et al. (2023) used endogenous fluorescent reporters for IRF1, CXCL10, and CXCL9 in RAW 264.7 macrophage-like cells to show that

pulsatile IFN- γ stimulation elicits gene-specific expression dynamics, which modeling suggests arising from gene-specific chromatin opening kinetics. We've included it in our discussion (Lines 596-601) as a source of cell-to-cell variability that may conceptually direct future studies.

2. Do graphs of input pulses (such as 1D, 2A etc) represent the robot's inputs or a quantification of the dye+cytokine in the chamber? One might expect it takes some seconds (minutes?) for the cytokine to leave the chamber. The authors include a dye with the medium and cytokine to enable monitoring of cytokine in the chamber. This could be imaged/quantified fairly straightforwardly. Given the paper's focus on signal decay, knowing the decay of color in the cell culture channel would aid interpretations of results. The authors should provide quantification of dye intensity in the cell culture channel in place of, or in addition to, input figures such as 1D and 2A.

The input pulses in figures 1D and 2A represent a schematic of the IL-1 concentration for visualization purposes. We have updated the figure legends to make it clear it is a schematic. We also included new data from a high frequency imaging experiment (New Fig. S1 c, and d). The data demonstrates reproducible pulse profiles, and invariant AUCs of fluorescence between a 1X6 and 4X1.5-minute ensuring comparable amounts of overall cytokine stimulation. Please also see the next response.

3. The unintuitive results of Figure 2A could be explained entirely by the cytokine not being instantly removed and instead having a very small delay in "washing out" after each pulse. As the cells behave as if more pulses give you increased area under the curve despite the authors claim of AUC-preserving inputs. Plotting dye intensity in the cell chamber would resolve this potential alternative explanation. Imperfect switching off of the input would also explain why fewer pulses or longer gaps reduce the effect.

To directly test whether incomplete washout could explain the effects seen in Figure 2A, we performed an experiment in which we imaged dye intensity in the microfluidic chip every 5 seconds using the same 4X1.5-minute and 1X6-minute pulse protocols from live-cell experiments. This enabled us to capture a higher-resolution trajectory of the stimulus dynamics delivered to the cells. As shown in the new figure, while there is a brief washout phase after each pulse, the total area under the curve (AUC) remains consistent between the 1X6 and 4X1.5-minute conditions. These measurements were highly reproducible across replicates, confirming that the multi-pulse protocol delivers an AUC-matched input. Therefore, the observed differences in cellular response cannot be attributed to cumulative differences in cytokine exposure due to imperfect washout. We updated Line 234-239 to highlight this result and added new figures to the supplementary figure 1 (Fig S1c, and d).

4. The physiological relevance of extremely short pulse stimuli is unclear and somewhat overstated in the article. The manuscript states that in vivo TNF and IL-1 are cleared within minutes. The references indicate approximately 20 minutes for TNF and 3 minutes to distribute and 41 minutes to eliminate IL-1Beta. Other studies give peaks at 10 minutes but sustained presence of stimuli in circulation and tissues for hours. (Newton RC, et al. Lymphokine Res. 1988). The study of Oyler-Yaniv used to further justify the relevance of short pulses used 5h pulses. While the experimental set up is

interesting to probe the extreme limits of the signalling axis, the physiological relevance is tenuous. Please re-write to remove this justification and instead focus on the study as testing the limits of NF-kB signalling.

We agree with the reviewer that physiological relevance is difficult to establish based on our current knowledge of cytokine dynamics *in situ*. We have re-written the related paragraph, highlighting some of the IL-1 specific literature that describes pulses or waves associated with secretion patterns and pyroptosis in cell communities (lines 214-220). In the context of the biology in our manuscript, we found this evidence to be more compelling and relevant, finishing the paragraph with a broader statement of interest in probing the limits of signaling as suggested by the reviewer (lines 224-226). We hope the reviewer will agree that the revised paragraph is a significant improvement.

5. Figure 2C is presented to indicate a loss of correlation between NEMO and RelA (compared to 1F). However, the control condition (no stimuli) is only included in figure 1 and appears essential for the correlation. Control conditions should either be included or excluded in both Figure 1 and Figure 2C.

We thank the reviewer for pointing this out, and we do agree that the unstimulated condition can artificially 'ground' the data correlation. To address this concern, instead of 'spiking in' the control data figure 2c, we opted to remove the control data in the scatter plot of Fig. 1f and in the calculation of the spearman correlation which was reduced ρ from 0.85 (including the control) to 0.76 (excluding the control). This is still a strong correlation compared to the weak correlation between the multi-pulse data in figure 2C, which is reported as 0.35. We believe this has strengthened our result.

6. The authors reveal t-adapt is key. It seems that for pulsatile stimuli the time that the final input ends is fairly predictive of the time that NEMO spots reach basal levels, and therefore AUC of RelA. The results could be summarised as: while NEMO is > 5 spots RelA decays with zero order kinetics, else it declines with first-order kinetics. Put simply perhaps a more simple explanation than the one explored by the paper is that if there are any C1 assemblies, there is sufficient NEMO for slow NF-kB decay. This explanation doesn't require epigenetics, therefore weakening the motivation for the rest of the paper.

An interesting observation and we believe our optimized D2FC model has already tested this hypothesis. Both of our optimized D2FC and D2FC² were simulated the activation of NF- kB by directly inputting our experimental time course trajectories. If the zero order dynamics were simply dependent upon having >5 NEMO spots, we believe the optimized D2FC model would have been able to convert from first order to zero order kinetics (Supplementary Figure S5d-g).

To further test the alternative hypothesis, we re-calculated our Rel-A exit rate scores where we aligned the starting point for fitting nuclear RelA exit rate dynamics with the IKK adaptation time. Using this approach resulted in fitting an exit rate curve at a later timepoint. If the nuclear RelA exit rate was dependent upon the IKK spots, then we should see a significant reduction in the number of nuclear RelA exit rate dynamics showing zero order dynamics. Although we do see a reduction in the fraction of zero order trajectories, the number of single cell trajectories showing

zero order remains consistently higher than the 1X6 minute pulse. Therefore, we believe the zero order exit rate dynamics depends on other critical regulatory components. We clarified our reasoning in why the D2FC model recapitulates the observed dynamics in lines 347-350 to clarify this point in our main text.

7. The D2Fc model used for particle swarm optimization does not contain many negative feedback loops that may impact decay of nuclear NF- κ B = I κ Bbeta, epsilon, p100/I κ B delta. This is particularly important given the conclusion of missing topology. While adding additional topology, focused around DNA binding, improved model fits, no other model architecture was investigated. Could the authors investigate whether adding additional feedback loops with different timescales/kinetics could explain the data? For example p100/I κ Bd and I κ Be.

We thank the reviewer for suggesting to test additional feedback topologies. In response to we investigated the contribution from other feedback topologies that have well defined mechanisms and associated models in new supplementary figure S7. By investigating feedback topologies reported by Kearns et al. (2006), Werner et al. (2005), and Hoffmann et al. (2002) we found that none of them could recapitulate the emergent property and the D2FC² significantly outperformed them all by a large margin. In combination with our additional evidence supporting the emergence of new transcriptional start sites (through H3K4me3 staining) and NF- κ B binding sites (through FRAP experiments), we feel that exploring these additional well-characterized feedback pathways strengthen our results. We also updated the text in lines 379-385 mentioning the new model results.

8. The only experimental validation of chromatin accessibility being the correct mechanism to explain the emergent behavior of cells is an experiment in which azacitidine causes increased RelA activity and decreased RelA induction. Due to the quantification of RelA induction as fold changes, it is somewhat inevitable that increased nuclear activity at basal will result in reduced fold changes. Therefore, the main prediction is increased RelA at basal. In a different cell line Azacitidine has been

previously shown to inhibit nuclear RelA at 1, 2 and 3 days, the authors find no change at 3 days and an increase at 6 days (Khong et al., 2008, Haematologica) suggesting the results presented here are cell-type specific rather than universally applicable.

Khong et al. treated the multiple myeloma (MM) cell lines U266 and H929 with 5 μ M azacitidine and observed a decrease in nuclear NF- κ B levels using a NoShift Transcription Factor assay kit. While we do not dispute their findings, it is important to note that MM cell lines are well characterized as having constitutive NF- κ B activation (PMID: 38514625), indicating that the basal levels of NF- κ B in these cells are likely to be substantially higher than expectations from U2OS cells, where we observe only low levels of basal nuclear NF- κ B. With constitutive NF- κ B, the MM cells most likely have a different chromatin landscape when compared to our U2OS cells and could explain why there are differences in our observations. Nevertheless, we have removed the Azacitidine data based on comments here and those from Reviewer 1.

9. An alternative hypothesis is that RelA is induced through another mechanism. For example, cell death in the cell culture which occurs in U2OS cells treated with DNA methylation inhibitors (Al-Romaih et al., cancer Cell International 2007).

Al-Romaih et al. treated U2OS cells with 1 μ M decitabine for five days and observed an increase in doubling time from 66 hours (control) to 84 hours in the treated cells. They also reported an increase in cell death, as measured by PI staining, from 5% in controls to 15% following decitabine treatment. These results suggest that a subpopulation with \sim 10% of cells is highly susceptible to cell death, and that apoptosis is likely not the primary mechanism of action. In addition, Al-Romaih et al. demonstrated that DNA demethylation occurred by three days of decitabine treatment. Although we did not measure cell death directly, our cells showed no visible signs of stress under the microscope after five days of decitabine treatment compared to the DMSO control. Additionally, cells treated with decitabine exhibited increased basal nuclear RelA levels, consistent with observations from azacitidine-treated cells, thereby providing an independent hypomethylating agent to support our hypothesis. However, given the strong experimental support from our FRAP and H3K4me3 analyses, we chose not to include decitabine and removed other hypomethylating agents in our final validation experiments.

Reviewer Figure 7. U2OS double-CRISPR cells were treated with 1 μ M decitabine for 5 days, after which the initial nuclear fluorescence intensity of mCh-RelA was quantified. A statistically significant increase was observed compared to the control condition (Student's t-test)."

10. I would have thought that if nuclear retention on DNA is key to the observed dynamics then this model would decouple NEMO and RelA dynamics, such that increased nuclear RelA is present without increased NEMO. Is that seen in this model? Is that seen in the data in response to azacytidine. Without this test there are too many alternative hypotheses that explain the data. Show this data.

We interpret your hypothesis that, if NEMO dynamics are similar between azacytidine-treated and control cells, then the azacytidine-treated cells should exhibit higher nuclear RelA levels. This would imply that chromatin remodeling enhances nuclear retention of RelA independently of upstream IKK/NEMO signaling. This restates our original hypothesis based on intuition alone (see also reviewer 1 comment 3): in azacytidine-treated cells, we anticipated that a single 6-minute pulse would lead to a zero-order nuclear export dynamic of RelA, reflecting sustained nuclear retention. However, azacytidine-treated cells exhibited increased basal levels of nuclear RelA and a reduced fold change in response to stimulation. These results were consistent with our model predictions, which showed that increasing chromatin accessibility leads to higher baseline nuclear RelA and dampened dynamic responses. Although the azacytidine data were consistent with our model predictions, we decided to remove them in favor of stronger, more direct model validation data obtained from other experiments (new figures 6 and 7). We believe the results are much more intuitive and direct.

Minor

11. Line 57-60 suggest that stochasticity is the only source of variability. There is substantial variability can be explained deterministically with sufficient characterisation (Roy et al., 2018, PNAS).

In Roy et al. (PMID: 29735717), the authors demonstrated that manipulating growth patterns through physical confinement can predictably induce nuclear reprogramming. This highlights how a population of cells can respond differently to external stimuli. Our intention in the original statement on lines 57–60 was to emphasize the challenges of studying signal transduction in a clonal cell population, where exposure to identical environmental stimuli still produces heterogeneous responses, despite no expected cell differentiation. We have updated the text (lines [57-58]) to clarify the type of variability being addressed.

12. Line 88 suggests only RelA fusion is available but recently reported cRel fusion proteins should be mentioned (Rahman et al., 2024, Cell Reports)

We updated the text to include the reference, lines 96-99.

13. Consider adding a fitting line to Fig 1F

We decided to not include a fitting line because the statistic we used for correlation was a Spearman rank correlation which does not assume linearity of the variables but rather assess a monotonic relationship between the two variables.

14. Ensure the Greek kappa is used throughout

Thank you for catching this inconsistency. We updated all instances of NF-kB to NF-κB. We did not use the Greek kappa for the model equations for consistently between the code and text.

15. Why U2OS cells here? A lot of previous data in MEFs is not comparable to the data generated here as a result of the different cell system. Similarly many existing models have been fitted to MEF data but are not used here. Why?

The foundational study for cytokine-dependent IKK activation by polyubiquitin chains was performed in U2OS cells by Xu et al. in 2009 (PMID: 19854138) and continued in our lab (PMID: 34301608), as well as other labs e.g. Tarantino et al. (PMID: 24446482), Du et al. (PMID: 35477005), etc... In addition to these studies in U2OS, we and others have been working with the pathway in human cancer cell lines for decades. Together, these studies have had a diversifying effect on our understanding of cytokine signaling and the underlying molecular circuits that would not likely have been found by MEFs alone. From our experiences, comparing dynamics in the pathway between human cell lines have been consistently like each other, and consistently different from results in MEFs. For the ubiquitin-specific mechanisms in U2OS, we have added additional justification on lines 115-116.

Dear Reviewers:

We would like to thank you again for your insightful and highly constructive comments. Our efforts to address the remaining concerns from the second submission are described below in the point-by-point response. These are primarily textual changes and include 1 additional experiment using the MLL1 histone methyltransferase inhibitor OICR-9429, shown in Figure 7C and Fig. S9C. Through this data, we were excited to find through chemical inhibition that MLL1 is the primary histone methyltransferase that cooperates with NF- κ B and promotes enhanced H3K4me3 following stimulation with a cytokine pulse train. We hope the reviewers will similarly be excited by this additional mechanistic demonstration and find our revised manuscript acceptable for publication in Nature Communications.

As previously, in the revised manuscript textual changes are marked in blue font.

Reviewer #1 (Remarks to the Author):

During the first review, we emphasized the need to directly demonstrate that DNA-mediated sequestration of NF- κ B underlies the switch in its nuclear-export kinetics. In the authors responses, they combined FRAP and HeK4me3 immunofluorescence to address our major concern. Figure 6 shows that pulsatile IL-1 stimulation increases the immobilized NF- κ B through FRAP, indicating increased NF- κ B binding. Figure 7 reveals a parallel rise in H3K4me3, a modification that opens chromatin, under the same stimulation. These complementary data align with their proposed mechanisms, and strengthens the argument that the interaction with DNA modulate the NF- κ B dynamics.

We appreciate the authors' efforts to address our concerns. However, we think a key mechanistic link between the results of figure 6 and 7 is still missing. The observed increase in immobilized NF- κ B and H3K4me3 under pulsatile IL-1 stimulation are correlative. To establish the causality, we recommend the authors to directly reduce H3K4me3 by inhibiting relevant methyltransferases, for example SET1 or MLL1 through knockdown or small molecules such as MI-501 (PMID: 25817203) or OICR-9429 (PMID: 34154613). Through these interventions, the authors could examine whether the immobilization of NF- κ B will be reduced or the export kinetics under pulsatile IL-1 stimulation will be reversed.

We thank the reviewers and appreciate their continued attention and help. In the following response, we will explain why we do not fully agree with the interpretation of the proposed intervention, but why it is still a very useful suggestion. First, we highlight the work of Wang

et al. (JCS 2012; PMID: 22623725). In this work the authors demonstrate that the MLL1 complex co-immunoprecipitates with NF- κ B (both cytoplasmic and nuclear), and MLL1 inhibition significantly disrupts transcription, upon cytokine stimulation, of a subset of NF- κ B-regulated genes, specifically the negative feedback mediators I κ B α and TNFAIP3. Furthermore Wang et al. show that MLL1 disruption does not alter activation of NF- κ B nor its DNA-binding profile i.e. these results position the activity of MLL1 as a secondary event that follows NF κ B pioneering (a relatively common interpretation for many histone methyltransferases PMID:38958075) and is required for NF- κ B mediated regulation of some genes. Surprisingly, Wang et al. show increased H3K4me3 enrichment at the I κ B α gene locus following cytokine stimulation, despite being a prototypical early and primary response gene that is accessible and basally expressed before stimulation. Taken together, MLL dependent H3K4me3 is required for inducible expression of NF- κ B early response genes that include negative feedback mediators. Accordingly, MLL1 inhibition will certainly alter the nuclear dynamics of NF- κ B, but in a way like translation inhibition (PMID: 31446223 Fig. S9) where reduced protein expression for negative feedback mediators prevents nuclear NF- κ B export after the cytokine response. These dynamics lean towards the opposite of what we understand is proposed by the reviewer. NF- κ B dynamics in the presence of MLL1/HMT inhibition and other chromatin-targeting perturbations would be indirect, peripheral to the results, and subject to the same criticisms as the Azacytidine results that we removed in revision 1.

On the other hand, the results of Wang et al. offer another critical hypothesis. Since MLL1 activity is tightly interleaved with NF- κ B, then MLL1 should be specifically among the histone methyltransferases (HMTs) that contribute to enhanced H3K4me3 following IL-1 stimulation. So, we tested this by pre-conditioning cells on OICR-9429 before stimulation with a 4x1.5 pulse train of IL-1. Remarkably, in the presence of OICR-9429, H3K4me3 staining is reduced to baseline placing MLL1 as the central HMT in the process (additional box in Fig 7c boxplot). Also consistent with the Wang et al. results, we see minimal impact on the immobilized fraction of NF- κ B following a pulse train in the presence of OICR-9429 (Fig. S9c). Taken together, the requested experiment improves the data in Fig. 7 by demonstrating a mechanistic role for MLL1 as the methyltransferase that cooperates with NF- κ B and promotes enhanced H3K4me3 following stimulation with a cytokine pulse train. We hope the reviewers are similarly excited by the result.

In addition to the suggestions above for deepening mechanistic insight we have several suggestions to improve the data presentation.

Figure 6

1. In panel B, the authors showed an example image. It will be more convincing to show example images for both 1X6min and 4X1.5min stimulations.

Panel B of figure 6 is intended to demonstrate a typical FRAP experiment for readers who are un-familiar with the process. Directly comparing the FRAP experiments between the 1X6min and 4X1.5min visually does not clearly demonstrate the fluorescence recovery curves and total immobilized protein, which requires quantification. Rather these total-RelA recovery curves are quantified in Figure S9A and summarized in Figure 6D.

2. In panel C, the legend describes the traces as example trajectories, but the curves are unusually smooth. Please clarify whether any filtering or processing was applied?

Panel C shows schematic trajectories included for illustrative purposes only, to demonstrate how total immobilized NF- κ B was defined. These trajectories appear smooth because they are not experimental data but conceptual representations. The experimental data are in Figure S9A. We have updated the figure legend to highlight this point.

3. The total immobilized NF- κ B level in panel C (around 3 units) and panel D (up to 100 units) are quite different. Please explain the source of this discrepancy?

During figure preparation, the y-axis label indicating a multiplication factor of 10^2 was inadvertently omitted from this panel. We have corrected the figure to include this label, which improves the consistency between the schematic in Panel C and the experimental data shown in Panel D.

Minor point

1. The “Simulation of Chromatin Opening” section in the methodology still mentions azacitidine experiment, which has been removed from the manuscript. Please revise this section according to the new manuscript.

We thank the reviewer for careful reading of our manuscript and removed the sentence discussing the azacitidine simulation.

Reviewer #2 (Remarks to the Author):

The authors have adequately addressed my questions and I now recommend publication without further changes

Thank you, Reviewer 2, for your help and support.

Reviewer #3 (Remarks to the Author):

The authors have made strong efforts to answer my comments from the previous review.

The graphs from high frequency imaging experiments, and AUC-controlled data are compelling and greatly strengthen the study. The removal of the AZA data has also better focused the study on the most compelling data.

The authors admit that the drug-response data was not the strongest, and that the physiological relevance was understated. Therefore the impact of this study beyond this experimental system remains somewhat unclear and the overall readership of the manuscript now that it is focused on the key data may be somewhat smaller.

Despite this, I believe the manuscript makes a substantial contribution to the field and will be of interest to many reader in Nature communications.

Thank you, Reviewer 3, for your help and support.